



# An improved carbon greenhouse gas simulation in GEOS-Chem version 12.1.1

Beata Bukosa[1,2], Jenny A. Fisher[2], Nicholas M. Deutscher[2], and Dylan B. A. Jones[3]

[1]National Institute of Water and Atmospheric Research, Wellington, New Zealand
[2]Centre for Atmospheric Chemistry, School of Earth, Atmospheric and Life Sciences, Faculty of Science, Medicine and Health, University of Wollongong, NSW, Australia
[3]Department of Physics, University of Toronto, Toronto, ON, Canada

**Correspondence:** Beata Bukosa (beata.bukosa@niwa.co.nz)

**Abstract.** Understanding greenhouse gas–climate processes and feedbacks is a fundamental step in understanding climate variability and its links to greenhouse gas fluxes. Chemical transport models are the primary tool for linking greenhouse gas fluxes to their atmospheric abundances. Hence accurate simulations of greenhouse gases are essential. Here, we present a new simulation in the GEOS-Chem chemical transport model that couples the two main greenhouse gases: carbon dioxide

($CO_2$) and methane ($CH_4$), along with the indirect effects of carbon monoxide (CO), based on their chemistry. Our updates include the online calculation of the chemical production of CO from $CH_4$ and the online production of $CO_2$ from CO, both of which were handled offline in the previous versions of these simulations. We discuss differences between the offline (uncoupled) and online (coupled) calculation of the chemical terms and perform a sensitivity simulation to identify the impact of OH on the results. We compare our results with surface measurements from the NOAA Global Greenhouse Gas Reference

Network (NOAA GGGRN), total column measurements from the Total Carbon Column Observing Network (TCCON) and aircraft measurements from the Atmospheric Tomography Mission (ATom). Relative to the standard uncoupled simulation, our coupled results show better agreement with measurements. We use the remaining measurement-model differences to identify sources and sinks that are over or underestimated in the model. We find underestimated OH fields when calculating the $CH_4$ loss and CO production from $CH_4$. Biomass burning emissions and secondary production are underestimated for CO in the

Southern Hemisphere and we find enhanced anthropogenic sources in the Northern Hemisphere. We also find significantly stronger chemical production of $CO_2$ in tropical land regions, especially in the Amazon. The model-measurement differences also highlight biases in the calculation of $CH_4$ in the stratosphere and in vertical mixing that impacts all three gases.

## 1 Introduction

Accurate simulations of greenhouse gases are vital for climate predictions. Carbon dioxide ($CO_2$) and methane ($CH_4$) are the

two main anthropogenic greenhouse gases and have significant impact on our climate. Due to human activities, the atmospheric amounts of $CO_2$ and $CH_4$ have increased globally by 40% and 150%, respectively, since the industrial revolution (Stocker et al., 2014). Carbon monoxide (CO) is less abundant than $CO_2$ and $CH_4$; however, through its indirect effect on $CH_4$, ozone and $CO_2$ it can also have a climate impact (Shindell et al., 2005). Changes in the atmospheric amounts of these gases, driven by





changes of their sources and sinks, largely control our future climate, but uncertainties about these processes and their budgets

still remain (Bousquet et al., 2006; Duncan et al., 2007; Liu et al., 2017). All three carbon greenhouse gases are chemically dependent, and a change in one can have a great effect on the other.

In the GEOS-Chem model, widely used for carbon gas flux inversion and source attribution, each of these gases have their own stand-alone simulation, decoupled from one another. Previous studies have emphasized the importance of the inclusion of the 3-D chemical production of $CO_2$ from the collective oxidation of CO, $CH_4$ and non-methane volatile organic compounds

(NMVOCs) (Enting and Mansbridge, 1991; Suntharalingam et al., 2005), but this chemical production, together with the secondary production of CO from $CH_4$, is handled offline in the stand-alone carbon gas simulations of the GEOS-Chem model (Nassar et al., 2010; Wecht et al., 2014; Fisher et al., 2017). Here, we present a new simulation in GEOS-Chem that couples $CO_2$, $CH_4$ and CO, with an online calculation of their chemical production for a more accurate simulation of these gases.

The majority of $CH_4$, the second most important anthropogenic greenhouse gas, is removed from the troposphere through reaction with OH:

$$CH_4 + OH \rightarrow CH_3 + H_2O \tag{R1}$$

that eventually leads to the formation of CO after a series of intermediate steps (Jacob, 1999):

$$CHO + O_2 \rightarrow CO + HO_2 \tag{R2}$$

Both $CH_4$ and CO have a common sink in the atmosphere through reaction with OH. The role of CO in determining tropospheric OH indirectly affects the atmospheric burden of $CH_4$ (Isaksen and Hov, 1987). It is one of the principal sinks of OH along with $CH_4$. Through reaction with OH, CO can also lead to the chemical formation of $CO_2$ (Enting and Mansbridge, 1991; Suntharalingam et al., 2005):

$$CO + OH \rightarrow CO_2 + H \tag{R3}$$

Oxidation of both primary CO, from direct anthropogenic and biomass burning emissions, and secondary CO, as an intermediate in the oxidation of $CH_4$ and NMVOCs, leads to the formation of $CO_2$. $CO_2$ can also be produced from the oxidation of carboxy–peroxy radical ($RCO_3$) and alkenoid ozonolysis (reaction of ethene with ozone; $C_2H_4 + O_3$) (Folberth et al., 2005), but this is assumed to be only a minor contributor.

Aside from anthropogenic sources and outside source regions, the major source of CO is $CH_4$ oxidation by OH through

Reactions (R1) and (R2) and the intermediate reactions. Early studies found the yield of CO from $CH_4$ oxidation ranging from 0.70–1 (Logan et al., 1981; Tie et al., 1992; Manning et al.; Novelli et al., 1999; Bergamaschi et al., 2000; Duncan et al., 2007). The CO chemical production term is estimated to be 760–1086 Tg CO $yr^{-1}$ (Holloway et al., 2000; Bergamaschi et al., 2000; Arellano Jr. and Hess, 2006; Duncan et al., 2007; Zeng et al., 2015; Fisher et al., 2017) and represents more than half of the total CO source.

The reaction of CO with OH radicals represents its largest sink, removing 2478–2630 Tg CO $yr^{-1}$ (Bergamaschi et al., 2000; Pétron et al., 2004; Arellano Jr. and Hess, 2006; Nassar et al., 2010; Fisher et al., 2017). The total chemical $CO_2$ source



is estimated to be around 1.04–1.1 Pg C yr$^{-1}$ (Suntharalingam et al., 2005; Nassar et al., 2010), which is about 12% of the annual anthropogenic $CO_2$ source (9.4 Pg C yr$^{-1}$, averaged for 2008–2017) (Le Quéré et al., 2018). Around 90–94% of the $CO_2$ chemical production is from CO oxidation (Folberth et al., 2005; Ciais et al., 2008). In contrast to the majority of the

$CO_2$ sources that are emitted at the surface, $CO_2$ from oxidation of CO is produced throughout the atmosphere. Although significant efforts have been made to constrain the total budgets of $CO_2$, $CH_4$ and CO, discrepancies in the chemical terms between studies suggest that these terms are still subject to uncertainties that can impact our understanding of the total budgets.

In this study, we introduce a new simulation in the GEOS-Chem model that couples the chemistry of three carbon greenhouse gases, $CO_2$, $CH_4$ and CO. With the new coupled simulation, our update eliminates the previously offline handling of the

chemical production between these gases (Nassar et al., 2010; Fisher et al., 2017), enabling us to have (i) better estimates of the chemical terms and (ii) simultaneous and consistent simulations of $CO_2$, $CH_4$ and CO that can help when constraining their fluxes based on their covariation (Wang et al., 2009; Pandey et al., 2015; Bukosa et al., 2019). Moreover, the coupled simulation removes the need to run the individual simulations separately if interested in all three gases, and it requires fewer computational resources than running three independent simulations.

We first describe the method for the online calculation of the chemical production, and the difference between the uncoupled and coupled versions of these simulations (Sect. 2). We then compare the stand-alone simulations of all three gases with the coupled simulation. For both versions we analyse their annual budgets and the contribution of chemical production to the total amount of each gas (Sect. 3 and 4) as well as their global spatial and temporal variability (Sect. 5). Finally, we validate the new coupled simulations against global surface flask measurements at sites part of the NOAA Global Greenhouse Gas Reference

Network (NOAA GGGRN), column measurements from the Total Carbon Column Observing Network (TCCON) and aircraft in situ measurements from the Atmospheric Tomography Mission (ATom) (Sect. 6).

## 2   GEOS-Chem

The coupled $CO_2$, $CH_4$ and CO simulation is based on version 12.1.1 of the GEOS-Chem 3-D global chemical transport model. The meteorological inputs for GEOS-Chem come from the Modern-Era Retrospective analysis for Research and Applications,

Version 2 (MERRA2) reanalysis developed by the NASA Global Modelling and Assimilation Office (GMAO). The native horizontal resolution of MERRA2 is 0.5°x0.625°. We ran the simulations at 2°x2.5° horizontal resolution with 47 vertical levels from January 2005 through December 2017. We used 10 min as the transport and convection timestep and 20 min for the chemistry and emissions timestep. Both the uncoupled and coupled simulations were initialized with a 10 year spinup for $CO_2$ and $CH_4$ using 2005 as a base spinup year, while for CO the model was spun up for 6 months in 2005. The spinup was carried

out with the uncoupled v11-01 simulations described in Bukosa et al. (2019). Due to differences between emission inventories used in Bukosa et al. (2019) and here, we use the first simulation year (2005) as an additional spinup year for all three gases. The production and loss terms used by each simulation are shown in Table 1 with additional common emission fields in the Supplement, Table S1. For simulation periods that are outside of the specified inventory time range, the model re-used the data from the closest year.





**Table 1.** GEOS-Chem production ($P$) and loss ($L$) terms used for the uncoupled and coupled carbon gas simulations.

| | $CO_2$ | $CH_4$ | CO |
|---|---|---|---|
| *Fields used by both uncoupled and coupled simulations* | | | |
| Tropospheric OH sink | - | Archived fields[a,b,c] | Archived fields[c] |
| Stratospheric $CH_4$ loss | - | Archived fields[d] | - |
| Stratospheric CO | - | - | GMI[e] |
| $P(CO)_{NMVOC}$ | - | - | $P(CO)_{NMVOC} = P(CO) - P(CO)_{CH_4}$ |
| | - | - | offline, full chemistry[c] |
| *Uncoupled only* | | | |
| $L(CH_4)$ | - | - | full chemistry |
| *Time resolution* | - | - | *Monthly mean, 2009–2011 average* |
| $P(CO)_{CH_4}$ | - | - | offline, $P(CO)_{CH_4} = L(CH_4)$ |
| *Time resolution* | - | - | *Monthly mean, 2009–2011 average* |
| $L(CO)$ | full chemistry | - | - |
| *Time resolution* | *Monthly mean, 2004–2009* | - | - |
| $P(CO_2)$ | offline, $P(CO_2) = L(CO)$ | - | - |
| *Time resolution* | *Monthly mean, 2004–2009* | - | - |
| *Coupled only* | | | |
| $L(CH_4)$ | - | - | online, from $CH_4$ |
| *Time resolution* | - | - | *Every model timestep, 20 min* |
| $P(CO)_{CH_4}$ | - | - | online, $P(CO)_{CH_4} = L(CH_4)$ |
| *Time resolution* | - | - | *Every model timestep, 20 min* |
| $L(CO)$ | online, from CO | - | - |
| *Time resolution* | *Every model timestep, 20 min* | - | - |
| $P(CO_2)$ | online, $P(CO_2) = L(CO)$ | - | - |
| *Time resolution* | *Every model timestep, 20 min* | - | |

[a] We use two types of OH fields for the calculation of the $CH_4$ loss via OH as part of a sensitivity test described in Sect. 2.3., [b] Park et al., [c] Fisher et al. (2017), [d] Murray et al. (2012), [e] NASA Global Modeling Initiative model

## 2.1 Uncoupled GEOS-Chem simulations

The existing uncoupled simulations are based on Nassar et al. (2010) and Nassar et al. (2013) for $CO_2$, Wecht et al. (2014) and Maasakkers et al. (2019) for $CH_4$ and Fisher et al. (2017) for CO.

These simulations are decoupled from other gases, hence they require input fields including chemical production rates and OH losses. GEOS-Chem can also perform a full chemistry simulation, known as the *coupled aerosol–oxidant chemistry in the*





*troposphere and stratosphere* simulation. The full chemistry simulation is required for the functionality of some of the stand-alone simulations because it provides input fields for those simulations. Various versions of the full chemistry simulation have been run previously to archive the production rates and oxidant fields used in the carbon gas simulations. Both the production and oxidant fields are computed using 3-D archives of monthly average values. All three carbon greenhouse gas simulations are linear, and each includes a suite of tracers tagged by source type and/or region.

The stand-alone $CH_4$ simulation in the troposphere is based on Eq. (1):

$$\frac{d[CH_{4Trop}]}{dt} = E_{CH_4} - S_{CH_4} - k_{CH_4,OH}[OH][CH_4] - \\ - k_{CH_4,Cl}[Cl][CH_4] \tag{1}$$

where $E_{CH_4}$ represents the surface emissions (gas, oil, coal, livestock, landfills, wastewater, biofuel, rice, biomass burning, wetlands, seeps, termites and other anthropogenic emissions), $S_{CH_4}$ is the sink from soil absorption, [OH], [Cl] and [$CH_4$] are the atmospheric concentrations of OH, Cl and $CH_4$, and $k_{CH_4,OH}$ and $k_{CH_4,Cl}$ are the pressure- and temperature-dependent rate constants for oxidation of $CH_4$ by OH and Cl, respectively. In the stratosphere, Eq. (1) becomes:

$$\frac{d[CH_{4Strat}]}{dt} = E_{CH_4} - CH_{4loss} \tag{2}$$

where $CH_{4loss}$ represents the stratospheric $CH_4$ sink based on stratospheric $CH_4$ loss frequencies archived from the NASA Global Modeling Initiative model (Considine et al., 2008; Allen et al., 2010) as described by Murray et al. (2012).

The CO simulation in the troposphere is based on Eq. (3):

$$\frac{d[CO_{Trop}]}{dt} = E_{CO} + P(CO) - k_{CO}[OH][CO] \tag{3}$$

where $E_{CO}$ represents the surface emissions (fossil fuel, biofuel and biomass burning), $P(CO)$ accounts for the chemical production of CO from $CH_4$ and NMVOC oxidation, [OH] represents the OH concentrations, and $k_{CO}$ is the pressure- and temperature-dependent rate constant for oxidation of CO by OH from the Jet Propulsion Laboratory (JPL) data evaluation (Burkholder et al., 2015). The chemical production of CO ($P(CO)$) can be further separated into the production from $CH_4$ ($P(CO)_{CH_4}$) and the production from NMVOC ($P(CO)_{NMVOC}$):

$$P(CO) = P(CO)_{CH_4} + P(CO)_{NMVOC} \tag{4}$$

The $P(CO)_{CH_4}$ and $P(CO)_{NMVOC}$ terms are obtained with the GEOS-Chem full chemistry simulation from the simulated monthly CO chemical production rates ($P(CO)$) as described by Fisher et al. (2017). In brief, the simulated $P(CO)$ is split offline to the $P(CO)_{CH_4}$ and $P(CO)_{NMVOC}$ terms based on the $CH_4$ loss rates ($L(CH_4)$) that are also simulated and saved from a full chemistry simulation. A 100% CO yield from $CH_4$ is assumed, hence the production of CO from $CH_4$ is equal to the $CH_4$ loss:

$$P(CO)_{CH_4} = L(CH_4) \tag{5}$$





The remaining $P(\text{CO})_{\text{NMVOC}}$ contribution is then calculated as the difference between the total CO production and the production of CO from $CH_4$:

$$P(\text{CO})_{\text{NMVOC}} = P(\text{CO}) - P(\text{CO})_{\text{CH}_4} \qquad (6)$$

Since the 100% yield may overestimate the production of CO from the oxidation of $CH_4$ the simulation caps the $P(\text{CO})_{\text{CH}_4}$ to the total $P(\text{CO})$ where it is greater than $P(\text{CO})$ (Fisher et al., 2017). In the stratosphere Eq. (3) becomes:

$$\frac{d[\text{CO}_{Strat}]}{dt} = \text{CO}_{prod} - \text{CO}_{loss} \qquad (7)$$

where $\text{CO}_{prod}$ represents the stratospheric production of CO from $CH_4$, while $\text{CO}_{loss}$ is the stratospheric CO sink due to chemical reaction with OH. Both quantities are from the NASA Global Modeling Initiative model.

The stand-alone $CO_2$ simulation throughout the atmosphere is based on:

$$\frac{d[\text{CO}_2]}{dt} = E_{\text{CO}_2} + P(\text{CO}_2) + D_{\text{CO}_2} \qquad (8)$$

where $E_{\text{CO}_2}$ represents the surface (fossil fuel, biomass burning, biofuel, shipping) and 3-D (aviation) emissions, $P(\text{CO}_2)$ accounts for the 3-D chemical production from the oxidation of CO, and $D_{\text{CO}_2}$ represents the net source from ocean exchange, balanced and net annual terrestrial exchange. Note that $D_{\text{CO}_2}$ can be positive or negative since these processes have negative values in regions where they act as a net sink and positive values where they act as a net source. The chemical source $P(\text{CO}_2)$ is a prescribed field that is calculated offline and read in at the start of the simulation. This chemical source is based on the monthly CO loss rates ($L(\text{CO})$) from the GEOS-Chem $4°\text{x}5°$ full chemistry simulation (Nassar et al., 2010). It is assumed that the $CO_2$ production is equal to CO loss:

$$P(\text{CO}_2) = L(\text{CO}) \qquad (9)$$

Some of the emission inventories used in the $CO_2$ simulation already include $CO_2$ from CO oxidation (effectively assuming prompt oxidation of precursors at the point of emission), but these amounts are only in the form of surface emissions, rather than distributed throughout the atmosphere, leading to a bias in the model (Suntharalingam et al., 2005). With the inclusion of a 3-D chemical source in the $CO_2$ simulation this bias needs to be corrected by subtracting the $CO_2$ chemical production "emitted" at the surface (in the emission inventories) from the total $CO_2$. Nassar et al. (2010) quantified a 0.825 Pg C $\text{yr}^{-1}$ global annual value for this surface correction based on emissions of all reactants that undergo oxidation to $CO_2$ and are included in emission inventories. This includes emissions from fossil fuel, biospheric $CH_4$ (wetlands, ruminants, rice, termites, landfill) and biospheric NMVOC emissions (isoprene and monoterpene). The emission inventories used for biofuel and biomass burning explicitly accounted for $CO_2$, CO, $CH_4$ and NMVOC separately, hence no surface correction was applied.

## 2.2 Coupled GEOS-Chem simulation

Our updates couple $CO_2$, $CH_4$ and CO based on the chemical loss and production between these species.





The starting point of the coupled simulation is the calculation of $CH_4$ based on Eq. (1) and (2). First, however, we made a minor correction to the treatment of $CH_4$ loss. In the uncoupled $CH_4$ simulation, the diurnal variability of OH (based on monthly mean values) was neglected, which would overestimate $CH_4$ loss at night and underestimate it during the day, with biases of up to 100% in some regions. In other offline GEOS-Chem simulations including CO, OH concentration is scaled to the cosine of the solar zenith angle to approximate diurnal variability. We updated the $CH_4$ code to also apply the diurnal scaling to the OH field used for $CH_4$ oxidation in the troposphere. Using a 1-year test simulation for 2005, we found that this change has a negligible impact on global annual tropospheric $CH_4$ loss. The calculation of the $CH_4$ loss in the stratosphere is unchanged between the coupled and uncoupled simulations. Due to the negligible difference in the annual $CH_4$ values with (in the coupled simulation) and without (in the uncoupled simulation) the diurnal cycling we have not run the uncoupled $CH_4$ simulation with this update for 2006–2017.

In the new coupled simulation, the tropospheric $CH_4$ loss rates are calculated from the oxidation of tropospheric $CH_4$ by OH at every time step. As before, a 100% yield of CO from $CH_4$ oxidation is assumed (Duncan et al., 2007), and the tropospheric $CH_4$ loss is passed to the CO part of the simulation at every timestep as the chemical production of CO from $CH_4$ ($P(CO)_{CH_4}$) in the troposphere. The calculation of the CO production in the stratosphere and from NMVOCs uses the same method as in the uncoupled CO-only simulation. In the troposphere, the total chemical production of CO ($P(CO)$) is equal to the sum of the archived $P(CO)_{NMVOC}$ field and the now online calculated $P(CO)_{CH_4}$.

The chemical production of $CO_2$ ($P(CO_2)$) is then calculated from the simulated CO loss from the oxidation of CO by OH in the troposphere and from archived CO loss in the stratosphere. As in the uncoupled version, a 100% yield of $CO_2$ from CO is assumed (Nassar et al., 2010). For the chemical surface correction, due to the inclusion of the chemically produced $CO_2$ in other emission inventories, we retain the same correction method and values as in the uncoupled simulation (Nassar et al., 2010).

The new coupling now allows time-specific changes and tracking of the chemical production terms. This is an improvement to the uncoupled simulations where the prescribed fields were based on simulations of specific prior years, and therefore could not capture the year-specific variations and dependencies between these gases. In the uncoupled $CO_2$, $CH_4$ and CO simulations, all the prescribed chemical production and loss fields were monthly mean values, while with the coupled simulation these fields are now calculated online at every timestep (i.e., 20 min), allowing us to track the day-to-day and diurnal variability of the simulated chemical production terms. A schematic diagram of the coupling is shown in Fig. 1.

## 2.3 OH fields

The OH fields have a significant impact on the chemical production and loss terms. The uncoupled carbon gas simulations in the default v12 GEOS-Chem model all use OH fields saved from different versions of the full chemistry model: GEOS-Chem v5-07-08 for the $CH_4$ simulation (Park et al.), v9-01-03 for the CO simulation (Fisher et al., 2017), and v8-02-01 for the $CO_2$ simulation (Nassar et al., 2010).

In our coupled simulation, the more recent v9-01-03 OH fields are used for all aspects of the simulation. However, the loss calculations do not feed back to OH and this aspect remains uncoupled. To test the impact of the different OH fields on the

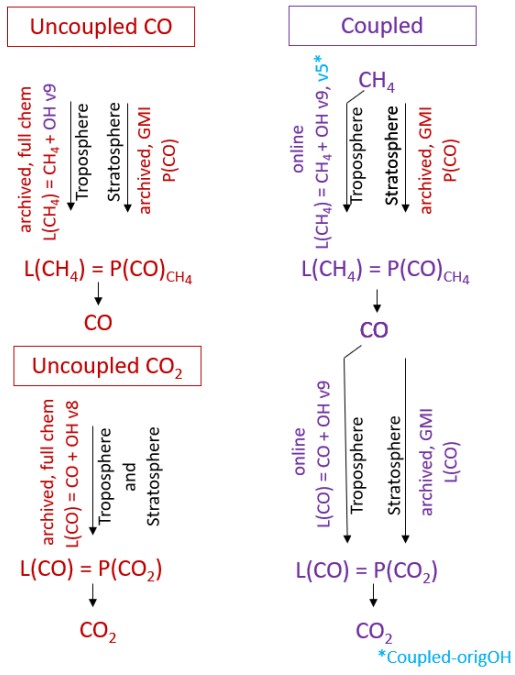

**Figure 1.** Schematic diagram of the uncoupled CO and $CO_2$ simulations (left) versus the coupled $CH_4$–CO–$CO_2$ simulation (right). The diagram also shows the OH field versions used between simulations, as described in the text (uncoupled CO: v9-01-03, uncoupled $CO_2$: v8-02-01, coupled: v9-01-03, *coupled-origOH*: v5-07-08 ($CH_4$ loss) and v9-01-03 (CO loss)). Colors correspond to simulations shown in subsequent sections (see text for details). Note, both simulations use the same $P(CO)_{NMVOC}$ field described in Sect. 2.1 (not shown on diagram).

chemical production and loss terms, we performed a sensitivity test where we retained the default version of the OH used in the $CH_4$ uncoupled simulation (i.e., v5-07-08 OH for the calculation of $L(CH_4)$). We will refer to this as the *coupled-origOH* simulation. A summary of the OH field versions is shown in Fig. 1.

The global annual mean OH is largest in the v8-02-01 full chemistry simulation ($11.8 \times 10^5$ molecules $cm^{-3}$) followed by v9-01-03 ($11.4 \times 10^5$ molecules $cm^{-3}$) and v5-07-08 ($10.8 \times 10^5$ molecules $cm^{-3}$) (http://wiki.seas.harvard.edu/geos-chem/index.php/Mean_OH_concentration). Figure 2 shows the yearly change of the OH fields used in the simulations. At the surface, v9-01-03 OH shows a peak during Northern Hemisphere (NH) summer (July), while v8-02-01 and v5-07-08 have an earlier peak in June; however, v5-07-08 also shows a second peak in October when the other two OH fields suggest a decline (Fig. 2).

The seasonal cycles at higher altitudes are more consistent between simulations.

Figure 3 shows the annual global spatial patterns of the OH fields. At the surface, the v9-01-03 OH fields are globally higher than v5-07-08 and lower than v8-02-01. Relative to v9-01-03, v5-07-08 shows lower surface OH above most land regions and NH ocean regions, while v8-02-01 only shows lower surface OH in tropical land regions and NH mid-latitude ocean regions, with higher OH elsewhere (Fig. 3). A similar pattern is observed at higher altitudes (500 hPa), with smaller and more



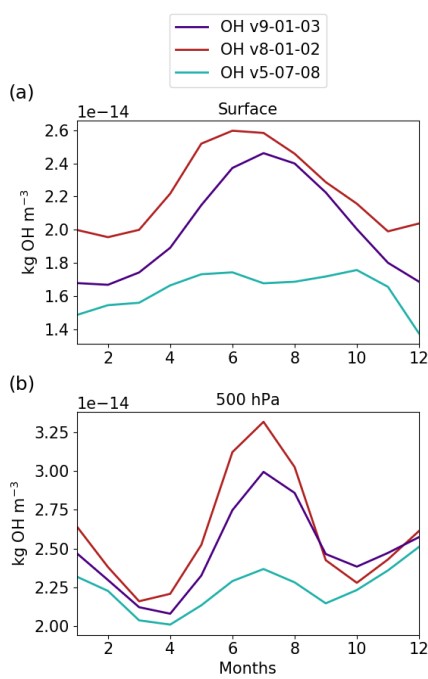

**Figure 2.** Globally averaged OH fields at the surface (a) and at 500 hPa (b) from the v9-01-03 (indigo, used by uncoupled CO and coupled simulation), v8-02-01 (red, uncoupled $CO_2$) and v5-07-08 (turquoise, uncoupled $CH_4$ and *coupled-origOH*) full chemistry simulations.

diffuse differences. The seasonal plots of the global surface distributions can be found in the Supplement, Fig. S2. Overall, we find significant differences between the OH fields used in the individual uncoupled simulations. These differences can have a large impact on the production and loss terms, as well the resulting mole fractions, which we test in what follows using the *coupled-origOH* sensitivity simulation. Going forward, we recommend that common OH fields should also be adopted for all uncoupled simulations in GEOS-Chem.

**3   Chemical production and loss budgets**

The main terms impacted by the coupling of $CH_4$, CO and $CO_2$ are the production of CO from $CH_4$ ($P(CO)_{CH_4}$) in the troposphere and the production of $CO_2$ from CO ($P(CO_2)$). Furthermore, the changes in these terms also impact the total source budgets for CO and $CO_2$ and the sink term for CO (loss of CO by OH ($L(CO)$)). Note, the tropospheric $CH_4$ loss by OH ($L(CH_{4Trop})$) is also different in the coupled simulation due to the inclusion of a diurnal OH cycle; however, as discussed

in Sect. 2.2 the impact of this update is negligible on the annual scale, and, as it is not a direct result of the coupling it is not further discussed here.

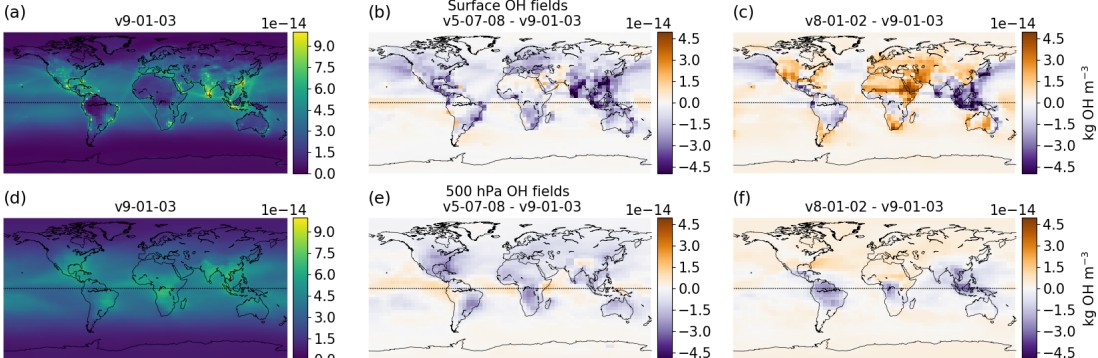

**Figure 3.** Surface (a–c) and 500 hPa (d–f) yearly averaged global spatial distribution of the OH fields based on the v9-01-03 (a, d) full chemistry simulation and the difference between v5-07-08 - v9-01-03 (b, e) and v8-02-01 - v9-01-03 (c, f).

We separate the analysis of our results into two parts: i) the impact of the coupling evaluated by comparing the uncoupled and coupled simulations and ii) the impact of using different OH (hereinafter referred to as the OH disconnect) evaluated by comparing the coupled simulation with the *coupled-origOH*.

### 3.1 Impact of the coupling

The annual global budgets of $P(CO)_{CH_4}$ and $P(CO_2)$ from the uncoupled and coupled simulations are shown in Fig. 4. The regional distribution of the annual production budgets, results from the *coupled-origOH* simulation and additional loss terms can be found in Fig. S3–S6.

We find an increase in $P(CO)_{CH_4}$ with time over the 2006–2017 period in the coupled simulation (Fig. 4b) due to increasing $CH_4$ mixing ratios, and hence increased $CH_4$ loss that is the most pronounced in tropical regions (Fig. S3). The $P(CO)_{CH_4}$ field in the uncoupled simulation (Fig. 4a) is based on 2009–2011 average values and therefore does not allow simulation of year-to-year changes. Different El Niño Southern Oscillation-triggered $CH_4$ processes lead to opposite changes in $CH_4$: during El Niño events, wetland emissions are reduced, while biomass burning emissions are enhanced (Dlugokencky et al., 2011; Hodson et al., 2011; Schaefer et al., 2018; Rowlinson et al., 2019). Our coupled simulation shows that these changes have important implications for the chemical production of CO that are not captured in the uncoupled simulation. We observe the strongest growth in the $P(CO)_{CH_4}$ during 2015/16, which coincidences with one of the strongest El Niño years, while we find no growth during 2010/11, a strong La Niña year, highlighting the impact of climate anomalies on the chemical terms. The availability of OH via CO also impacts the $CH_4$ interannual variability; however, we are unable to quantify the OH-driven changes here as none of our simulations include OH interannual variability or OH-feedbacks. We recommend future updates to the coupled simulation such as the prioritised inclusion of a $CO$–$OH$–$CH_4$ feedback in the calculation (Holmes, 2018).

Figure 5 shows the change of the budgets throughout the year for each chemical term in different latitudinal bands. Figure 5a shows that both the uncoupled (red) and coupled (indigo) $P(CO)_{CH_4}$ have a similar annual cycle, with overall stronger



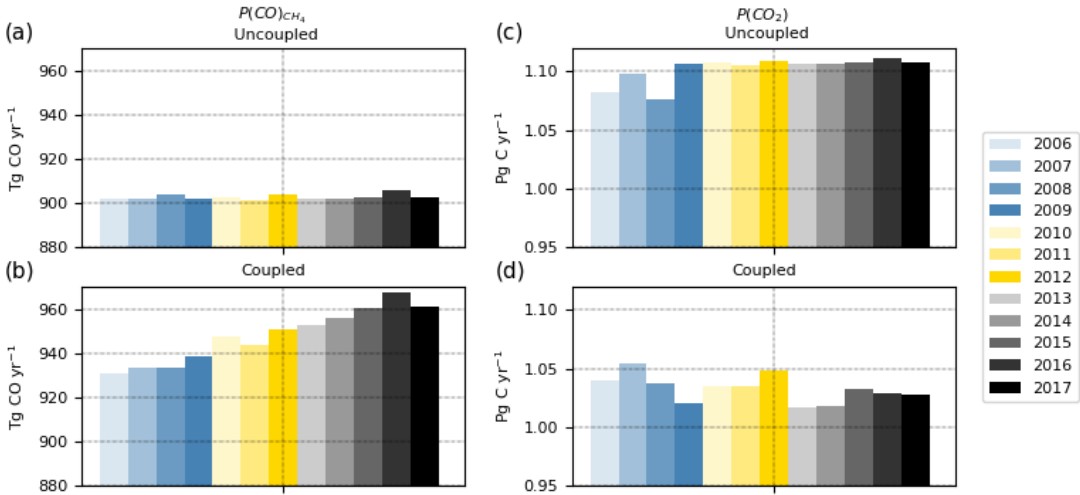

**Figure 4.** Annual values of the global chemical term budgets for CO production from $CH_4$ (a, b) and $CO_2$ production from CO (c, d) from the uncoupled (a, c) and coupled (b, d) simulations.

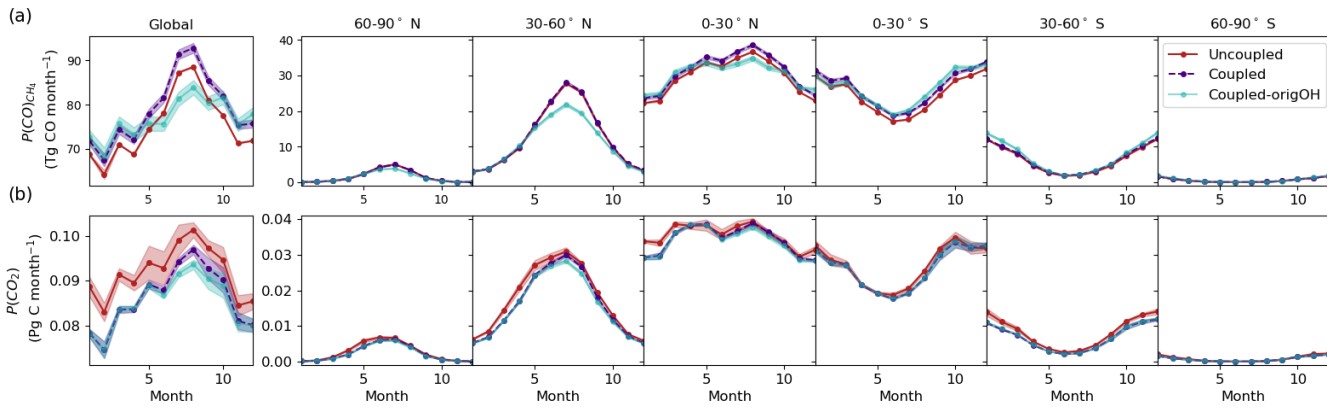

**Figure 5.** Monthly changes of the CO production from $CH_4$ (a) and $CO_2$ production from CO (b), with 1 standard deviation, from the uncoupled (red), coupled (indigo) and *coupled-origOH* (turquoise) simulation (averaged for 2006–2017).

production in the coupled simulation. The coupled simulation also shows stronger variability due to the year-specific $CH_4$ loss, with the strongest variability in tropical regions.

The $P(CO_2)$ in the uncoupled simulation has year-specific fields only for 2004–2009 (Nassar et al., 2010) (Fig. 4c); hence in contrast to the coupled simulation (Fig. 4d) it is missing information about the temporal change of this term after 2009. Although covering a shorter time period, from 2006 to 2009, the $P(CO_2)$ in the uncoupled simulation shows stronger vari-





**Table 2.** Global and regional budgets for $CH_4$ loss in the troposphere ($L(CH_{4Trop})$ in Tg $CH_4$ $yr^{-1}$), CO production from $CH_4$ ($P(CO)_{CH_4}$ in Tg CO $yr^{-1}$), CO loss from OH ($L(CO)$ in Tg CO $yr^{-1}$) and $CO_2$ production from CO ($P(CO_2)$ in Pg C $yr^{-1}$) from the uncoupled (U) and coupled (C) simulations, as well literature values for the global budgets. The budgets from the simulations are shown as a multi-year mean based on years 2006–2017. The range of values for individual years is shown in the parentheses.

| Chemical terms | Prior Work | Global | | NH | | SH | |
| --- | --- | --- | --- | --- | --- | --- | --- |
| | | U | C | U | C | U | C |
| $L(CH_{4Trop})$ | $382$–$617^{a,b,c}$ | $476^d$ | $510$ $(501$–$522)$ | $263^d$ | $294$ $(289$–$301)$ | $213^d$ | $216$ $(212$–$220)$ |
| $P(CO)_{CH_4}$ | $760$–$1086^{e,f,g,h,i}$ | $902^j$ $(901$–$905)^j$ | $947$ $(931$–$967)$ | $521^j$ $(520$–$522)^j$ | $542$ $(532$–$553)$ | $381^j$ $(380$–$382)^j$ | $405$ $(398$–$413)$ |
| $L(CO)$ | $2478$–$2630^{f,k,l,m}$ | $2363^j$ $(2320$–$2426)^j$ | $2408$ $(2370$–$2457)$ | $1438^j$ $(1417$–$1466)^j$ | $1461$ $(1446$–$1479)$ | $924^j$ $(900$–$962)^j$ | $946$ $(923$–$978)$ |
| $P(CO_2)$ | $1.04$–$1.1^{n,o}$ | $1.1^p$ $(1.08$–$1.11)^p$ | $1.03$ $(1.02$–$1.05)$ | $0.67^p$ $(0.63$–$0.68)^p$ | $0.63$ $(0.62$–$0.63)$ | $0.43^p$ $(0.43$–$0.46)^p$ | $0.41$ $(0.40$–$0.42)$ |

[a] Wuebbles and Hayhoe (2002), base year: only the "best guess" estimates are shown, based on a range of values, [b] Ciais et al. (2014), base year: 1980–1989, [c] Wang et al. (2004), base year: 1994, [d] Based on 2005 only, the 2005 values in the coupled simulation are Global: 501 Tg $CH_4$ $yr^{-1}$, NH: 289 Tg $CH_4$ $yr^{-1}$, SH: 211 Tg $CH_4$ $yr^{-1}$, [e] Holloway et al. (2000), base year: not defined, [f] Bergamaschi et al. (2000), base year: 1993–1995, [g] Duncan et al. (2007), base year: 1988–1997, [h] Arellano Jr. and Hess (2006), base year: 2000–2001, [i] Zeng et al. (2015), base year: 2004, range based on different model simulations. [j] Fisher et al. (2017), base year: 2009–2011 average, [k] Pétron et al. (2004), [l] Arellano Jr. and Hess (2006), [m] Nassar et al. (2010), base year: 2004–2010, these are the fields used in the uncoupled $CO_2$ simulation, 100% yield from $L(CO)$ loss to $P(CO_2)$, NH:1446–1558 Tg CO $yr^{-1}$, SH:1003–1075 Tg CO $yr^{-1}$, [n] Nassar et al. (2010), base year: 2000–2009, [o] Suntharalingam et al. (2005), base year: 1988–1997, [p] Nassar et al. (2010), base year: 2006–2009.

ability relative to the coupled one (Fig. 5b). The uncoupled simulation also shows stronger $P(CO_2)$ values in all latitudinal bands; however, the largest difference between simulations is during December–March mostly in NH tropical and Southern

Hemisphere (SH) mid-latitude regions. The stronger uncoupled $P(CO_2)$ values are a result of different CO amounts used for the CO loss calculation between the coupled and full chemistry simulations in GEOS-Chem, as well as more abundant OH used to calculate $L(CO)$ for the uncoupled simulation (v8-02-01, Fig. 2) relative to the OH field used in the coupled simulation (v9-01-03), discussed in Sect. 2.3.

The global and regional budgets for the chemical components for both coupled and uncoupled versions of the model along

with known literature values are shown in Table 2. The results from our coupled simulation are in good agreement with values from prior work. In summary, the coupled simulation shows stronger $P(CO)_{CH_4}$ than the uncoupled simulation due to the stronger $CH_4$ loss for all years and for both hemispheres (29–61 Tg CO $yr^{-1}$ difference). This difference represents 1.2–2.6% of the total CO source in the coupled simulation. The $CO_2$ chemical source shows weaker values in the coupled simulation relative to the uncoupled one (0.04–0.09 Pg C $yr^{-1}$ difference). This difference represents 0.3–0.7% of the total $CO_2$ source

in the coupled simulation.





## 3.2 Impact of the OH disconnect

As described in Sect. 2.3, we perform a sensitivity simulation (*coupled-origOH*) with the coupled simulation but using the default OH field from the uncoupled $CH_4$ simulation when calculating the chemical loss terms in order to highlight the impact of inconsistent OH fields on the simulated values. Using the default v5-07-08 OH fields for the $L(CH_{4Trop})$ calculation results in a 9–22 Tg $CH_4$ $yr^{-1}$ global decrease relative to the coupled simulation (Fig. S3), with weaker $CH_4$ loss in the *coupled-origOH* simulation due to the lower OH (Fig. 2). Relative to the standard coupled simulation, the *coupled-origOH* sensitivity simulation shows weaker $CH_4$ loss globally and in the NH, and stronger loss in the SH due to higher OH values over the SH ocean regions (Fig. S3 and Fig. 3).

Figure 5a shows that the weaker $CH_4$ loss in *coupled-origOH* also results in weaker global CO production from $CH_4$ (16–39 Tg CO $yr^{-1}$ difference). As with the $L(CH_{4Trop})$, the stronger coupled simulation values are only present in the NH, while in the SH the *coupled-origOH* results show stronger $P(CO)_{CH_4}$. As in the coupled simulation, the *coupled-origOH* simulation shows an increase in the $P(CO)_{CH_4}$ with time due to the increased $CH_4$ loss. Relative to the coupled results, the *coupled-origOH* $P(CO)_{CH_4}$ (Fig. 5a, turquoise line) globally shows stronger production between September–May and weaker production between June–August, with the largest difference in mid-latitude regions in the NH. These differences are exclusively driven by differences in the OH fields.

In contrast to the $P(CO)_{CH_4}$ values, using the v5-07-08 OH fields for the $L(CH_{4Trop})$ calculation has a negligible impact on $P(CO_2)$. This is expected since the $L(CH_{4Trop})$ has only an indirect and minor contribution when calculating $P(CO_2)$. Both the *coupled-origOH* and coupled simulations show similar $P(CO_2)$ budgets but with stronger production in the coupled simulation between July–October (Fig. 5b).

## 4 Chemical source contributions

Due to the linearity of the GEOS-Chem carbon greenhouse gas simulations, in addition to simulating the total amount of each gas, we can also quantify the mole fractions of individual processes (referred to as tracers). These include the $CO_2$ mole fraction from $CO_2$ chemical production ($CO_{2CO}$) and the CO mole fraction from CO production from $CH_4$ ($CO_{CH_4}$). Figure 6 shows these chemical production tracers (Fig. 6a, b), as well the total CO and $CO_2$ mole fractions (Fig. 6c, d) at the surface for different latitudinal bands. Note, in contrast to the CO source tracers where the atmospheric sink terms (e.g., OH) are applied to each tracer, for $CO_2$ there is no sink applied to the different source tracers. This leads to a trend in $CO_{2CO}$ and its accumulation in the atmosphere. To highlight differences in the seasonal cycle we have detrended the $CO_{2CO}$ data shown in Fig. 6b, d and added the mean 2006–2017 yearly growth rates.

Implementing the online calculation of the chemical terms results in higher $CO_{CH_4}$ values in the coupled simulation relative to the uncoupled along with stronger variability (Fig. 6a), similar to the production rates (Table 2, Fig. 5a). An average $1.2 \pm 0.5$ ppb difference is present across the NH between the coupled and uncoupled results while in the SH we find a larger difference of $1.8 \pm 0.5$ ppb. Both the coupled and uncoupled simulations show similar seasonal cycles; however, using inconsistent OH fields between simulations led to significant differences in the $CO_{CH_4}$ seasonal cycles. In the *coupled-origOH* simulation,



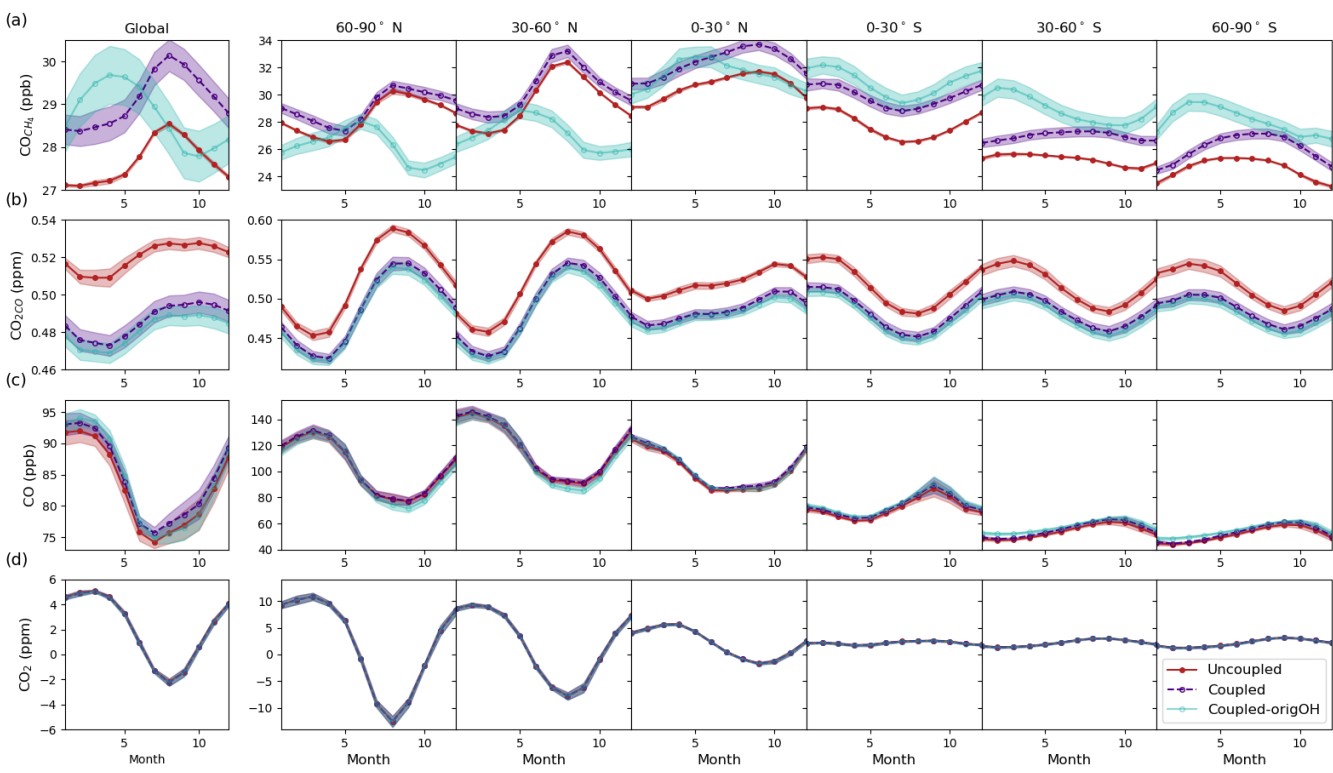

**Figure 6.** Surface mole fractions from chemical production of $CO_{CH_4}$ (a) and $CO_{2CO}$ (b) and total CO (c) and $CO_2$ (d) mole fractions from the uncoupled (red), coupled (indigo) and *coupled-origOH* (turquoise) simulations with 1 standard deviation (based on 2006–2017 average values). Note, the $CO_2$ values are detrended and added to the mean 2006–2017 yearly growth rates.

the mole fractions have a reversed seasonal cycle relative to both the coupled and uncoupled results, with the reversal most
pronounced in the NH mid-latitude and polar regions. A $2.3 \pm 2$ ppb difference is present in the NH between the coupled and *coupled-origOH* results, with higher values in the coupled simulation, while in the SH we find a $-1.7 \pm 1.1$ ppb difference, with higher values in the *coupled-origOH* simulation. The difference in the seasonal cycles is also reflected in the total amounts of CO (Fig. 6c); however, globally the total CO has the same seasonal cycle in all three simulations, since the contribution of $CH_4$ oxidation is only $\approx 35$ % of the total CO.

To understand the origin of the differences in the seasonal cycles we add an additional chemical term to the analysis, $L(CO_{CH_4})$ that represents the loss of $CO_{CH_4}$ via OH, which is a sub-component of the total $L(CO)$. The difference between the production ($P(CO)_{CH_4}$) and loss ($L(CO_{CH_4})$) together with transport leads to the modelled $CO_{CH_4}$ mole fractions. Figure 7 shows the global surface and 500 hPa altitude $P(CO)_{CH_4}$, $L(CO_{CH_4})$ and their difference from the uncoupled, coupled and *coupled-origOH* simulations and Fig. S7 shows the change of these terms in the NH versus SH. Both at the surface and in the



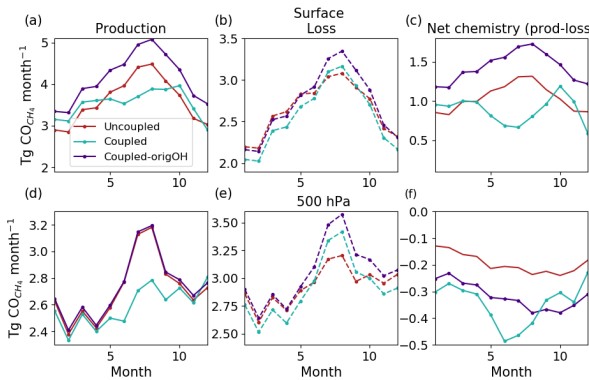

**Figure 7.** Global production of CO from $CH_4$ (a, d), its loss via OH (b, e) and their difference (c, f) in the uncoupled (red), coupled (indigo) and *coupled-origOH* (turquoise) simulations at the surface (a–c) and 500 hPa altitude (d–f), averaged for 2006–2017.

tropospheric column, the *coupled-origOH* simulation results show low production in June–August, a period when we observe the maximum production in both the uncoupled and coupled simulation, leading to an opposite seasonal cycle. As already discussed, $P(CO)_{CH_4}$ in the troposphere is calculated based on a 100% yield from the $CH_4$ loss via OH, hence the $P(CO)_{CH_4}$ and resulting $CO_{CH_4}$ mole fraction differences are a result of the differences in the $L(CH_{4Trop})$ fields. The main difference in the $CH_4$ loss calculation between the *coupled-origOH* versus the uncoupled and coupled simulation is the version of the OH

field. The v5-07-08 OH field used in the *coupled-origOH* simulation results in weaker $L(CH_{4Trop})$ during NH summer, which leads to weaker $P(CO)_{CH_4}$ and significant differences in the net chemistry (production-loss) and resulting mole fractions. We find that this difference is the most pronounced at the surface while at higher altitudes it slowly diminishes. Although further differences exist in the $L(CH_{4Trop})$ calculation between simulations, we find that the choice of the OH field is the dominant driver of the resulting discrepancies. The additional differences are described in the Supplement, Sect. S1.

The $CO_{2CO}$ mole fractions show a similar seasonal cycle between all three simulations in both hemispheres with a stronger seasonal cycle in the uncoupled simulation (i.e., stronger amplitude) in all regions except the NH tropics (Fig. 6b). The uncoupled simulation due to the stronger chemical production shows a stronger yearly global surface growth rate of 0.52 ppm $yr^{-1}$, followed by the coupled simulation, 0.49 ppm $yr^{-1}$, while the *coupled-origOH* shows a weaker growth rate of 0.48 ppm $yr^{-1}$ due to weaker production. Overall, both the coupling and OH disconnect do not significantly impact the resulting mole

fractions between simulations. Moreover, the differences between the coupled and uncoupled simulations are too small to be reflected in the total $CO_2$ surface values (Fig. 6d). As already highlighted $P(CO_2)$ is a 3-D source, hence the signal of this source in the surface mole fractions is small relative to the other more dominant $CO_2$ surface fluxes.

## 5  Global distribution

Figure 8 shows the total column chemical production of CO from $CH_4$ with corresponding mole fractions at the surface

and 500 hPa altitude from the three simulations, as well as their differences. $P(CO)_{CH_4}$ and $L(CH_4)$ have the same spatial


and temporal distribution due to the 100 % yield. Figure 9 is the same as Fig. 8 but for the chemical production of $CO_2$. Similarly, both $P(CO_2)$ and $L(CO)$ have the same spatial and temporal distribution due to the 100 % yield. For $CO_2$ we have additionally removed the long-term trend from the $CO_{2CO}$ mole fractions, as discussed in Sect. 4. The seasonal changes of both the productions and mole fractions are shown in Fig. S8–S11 for CO and Fig. S12–S15 for $CO_2$.

## 320   5.1   Impact of the coupling

The online calculation of $P(CO)_{CH_4}$ has a small impact on its global spatial distribution - both the coupled and uncoupled simulations show similar results (Fig 8a, b). The simulations use the same OH fields, hence the differences in the $P(CO)_{CH_4}$ are driven by different handling of the $CH_4$ values before the OH loss is applied (Supplement Sect. S1). The main difference between the two simulations is the stronger $P(CO)_{CH_4}$ over tropical ocean regions and weaker $P(CO)_{CH_4}$ over NH land

regions in the coupled version. $P(CO)_{CH_4}$ shows a seasonal cycle with maximum production during NH summer and minimum during winter, the opposite of the seasonal cycle of the total CO mole fractions (Fig. S12–S15). On a yearly scale, the surface $CO_{CH_4}$ mole fractions from the coupled simulation show higher values above both ocean and land regions (Fig 8i), as a result of the stronger $P(CO)_{CH_4}$ over tropical ocean regions. A similar behavior is observed at 500 hPa; however, the differences are smaller and more diffuse. We observe the same differences throughout the seasons. The seasonal change of the resulting mole

fractions depends on the change of the production and loss ratio (Sect. 4), as well the impact of transport, and can differ from the seasonal cycle of the production fields. We further discuss the simulated mole fractions and the impact of the coupling on total CO in Sect. 6.

The coupled simulation shows stronger $P(CO_2)$ (Fig 9b) in certain land regions, despite the annual global chemical source being weaker than in the uncoupled simulation by 0.04–0.1 Pg C yr$^{-1}$. South America, Central Africa, Indonesia, parts of East

Asia and Australia show stronger $CO_2$ chemical production relative to the uncoupled simulation. Moreover, in the uncoupled simulation there is almost no $P(CO_2)$ observed above the Amazon (Nassar et al., 2010); however, our results suggest the opposite. The difference patterns appear to be mostly independent of season (Fig. S12–S15). The chemical production is overall stronger above the ocean in the uncoupled simulation for all seasons; however, the coupled simulation does show stronger $P(CO_2)$ during certain periods in tropical and NH mid-latitude regions. The stronger $P(CO_2)$ above South America, Central

Africa, Indonesia, parts of East Asia and Australia shown in the yearly average fields are present in all seasons; but with the strongest contribution during September–November. South America, Central Africa and northern Australia are characterized by strong biomass burning, especially during the SH dry season when frequent fires are observed (September–November), emitting large amounts of CO into the atmosphere (Edwards et al., 2006). Our coupled model simulates the $P(CO_2)$ in these regions during the fire season to be stronger than the previous fields used in the uncoupled simulation. The stronger

$P(CO_2)$ from the coupled simulation in other regions such as East Asia and North America point to enhanced anthropogenic CO emissions that lead to stronger chemical production of $CO_2$. In addition to the primary CO emissions, the secondary production of CO from NMVOC could also have a significant impact on the $P(CO_2)$ in regions where we observe differences. Different model versions were used to save the $P(NMVOC)$ and $P(CO_2)$. The updated chemistry between model versions

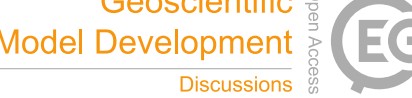

**Figure 8.** Average 2006–2017 total column CO chemical production from $CH_4$ (a–e), corresponding mole fractions (i.e., $CO_{CH_4}$) at the surface (f–j) and at 500 hPa (k–o) based on the uncoupled (a, f, k), coupled (b, g, l) and *coupled-origOH* (c, h, m) simulations, and their difference relative to the coupled simulation (d, e, i, j, n, o).

would additionally impact the $P(CO_2)$ through the CO production from NMVOC in regions where we expect a significant
contribution from this production term (e.g., the Amazon).



**Figure 9.** Average 2006–2017 total column $CO_2$ chemical production from CO (a–e), corresponding mole fractions (i.e., $CO_{2CO}$) at the surface (f–j) and at 500 hPa (k–o) based on the uncoupled (a, f, k), coupled (b, g, l) and *coupled-origOH* (c, h, m) simulations, and their difference relative to the coupled simulation (d, e, i, j, n, o).

The spatial distribution of the mole fractions is similar between simulations with overall higher values in the uncoupled simulation due to the globally stronger $P(CO_2)$, with higher mole fractions in the NH. However, the coupled simulation does



show more abundant $CO_{2CO}$ over land regions with stronger $P(CO_2)$, with larger differences at higher altitudes. The full chemistry simulation that was used to create the uncoupled simulation $P(CO_2)$ fields was run on lower resolution ($4°x5°$),

hence the uncoupled $P(CO_2)$ field is more diffuse relative to the coupled simulations.

### 5.2   Impact of the OH disconnect

Although we do not find significant differences in the $P(CO)_{CH_4}$ global spatial distribution between the coupled and uncoupled simulation, the *coupled-origOH* results show significant differences (Fig. 8c). We find stronger production over the ocean and weaker production above land regions in the *coupled-origOH* simulation relative to both the coupled and uncoupled

simulations. The $P(CO)_{CH_4}$ in the coupled simulation shows stronger values than in the *coupled-origOH* during all seasons, with the largest difference during summer periods for each hemisphere and larger differences in the NH; however, the difference over the ocean has a weaker seasonality. The $CO_{CH_4}$ mole fractions at the surface are highest in the tropical ocean regions in the *coupled-origOH* simulation due to the stronger chemical production in these regions. The *coupled-origOH* simulation also shows lower mole fractions in the NH and higher values in the SH; however, with lower mole fractions in parts of the SH,

Africa and South America. Due to a coarser resolution of the OH fields in the *coupled-origOH* ($4°x5°$) relative to the OH fields in the uncoupled and coupled simulations ($2°x2.5°$), the $L(CH_4)$ and $P(CO)_{CH_4}$ field in the *coupled-origOH* simulation is also more diffuse.

    The same OH field is used to calculate the $L(CO)$ and $P(CO_2)$ in the coupled and *coupled-origOH* simulation (Fig 9b, c), hence the $P(CO_2)$ is only impacted by differences in $P(CO)_{CH_4}$ through $L(CH_4)$ that relies on different OH fields. Both

simulations show similar spatial distribution, with stronger $P(CO_2)$ in the coupled simulation over land regions and in the NH. Differences in the mole fractions are also minimal; however, with higher values in the coupled simulation, especially in the NH.

### 5.3   Vertical latitudinal distribution

Figure 10 shows the vertical latitudinal distribution of $P(CO)_{CH_4}$ and $P(CO_2)$ for different months, averaged for 2006–2017,

as well as the differences between simulations relative to the coupled simulation.

    The strongest $P(CO)_{CH_4}$ in the coupled simulation occurs between the surface and 3 km altitude. For most months, this chemical production is stronger in the NH relative to the SH; however, around November we observe a stronger production in the SH, potentially due to biomass burning and wetland activity that leads to higher $CH_4$ levels and subsequent loss. Although the strongest production occurs between $50° S$–$50° N$, we also observe production in SH polar regions in November–January

and in Arctic regions in May–July, corresponding to their summer periods. The Arctic regions shows stronger production relative to the Antarctic regions, due to higher $CH_4$ levels and stronger loss in the NH. As for the global spatial distribution results, the coupling has a small impact on the vertical distribution (Figure 10b, stronger production in the coupled) while the OH disconnect shows larger differences (Figure 10c). The largest difference between the coupled and *coupled-origOH* simulation occurs in July–September, when the coupled simulation suggests stronger production at the surface in SH mid-

latitude regions and weaker production in the NH.



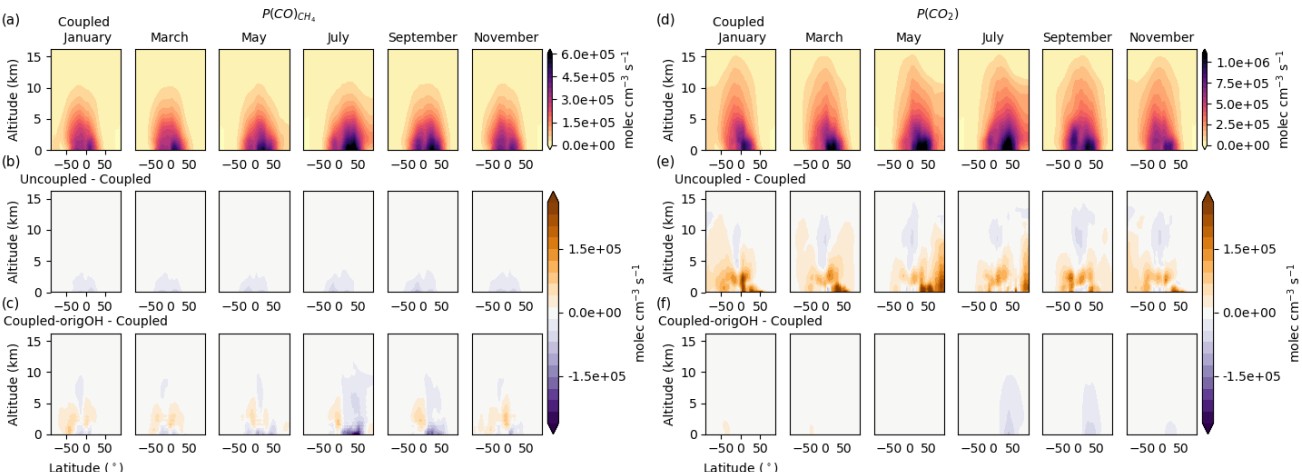

**Figure 10.** Vertical latitudinal distribution of the coupled CO chemical production from $CH_4$ (a) and $CO_2$ chemical production from CO (d) and the uncoupled (b, e) and *coupled-origOH* (c, f) differences relative to the coupled simulation, for different months, averaged for 2006–2017.

The strongest $CO_2$ chemical production in the coupled simulation occurs between the surface and 4 km altitude and $CO_2$ is produced chemically up to 15 km (Figure 10d). Between January–July we observe stronger production in the NH, with the strongest production in tropical regions at the beginning of the year, moving towards higher latitudes by July. Based on the distribution (Fig. 9) of this source in the NH, strong production occurs over China and India from anthropogenic

CO, with mixed biomass burning from other regions. For the remaining months, both hemispheres show strong $P(CO_2)$, with the SH showing stronger production in September, presumably due to additional biomass burning in the tropics (e.g., Indonesia, Australia, Africa, S America). Nassar et al. (2010), based on results from the uncoupled simulation, did not find a biomass burning contribution over the Amazon; however, our coupled simulation, as already discussed, suggests a significant contribution from this region. Relative to the uncoupled simulation, the coupled simulation shows weaker production in mid-

latitude and polar regions, with stronger contribution in the tropics at surface levels and above 5 km. The Arctic and Antarctic regions show weaker production in the coupled simulation. As for the global spatial distribution, the $P(CO_2)$ values between the coupled and *coupled-origOH* simulations are similar; however, with stronger production in the coupled.

## 6 Simulation comparison with measurements

We validate the new coupled simulation against global column retrievals, calibrated surface flask and aircraft in situ mea-

surements (Fig. 11, Table A1). Long-term time series of column averaged dry air mole fraction measurements of $CO_2$, $CH_4$ and CO are measured by TCCON (Wunch et al., 2011). In addition, long-term time series of surface mole fractions exist at different sites across the globe as part of NOAA GGGRN (Dlugokencky et al. (2020b, a); Petron et al. (2020)). For vertical

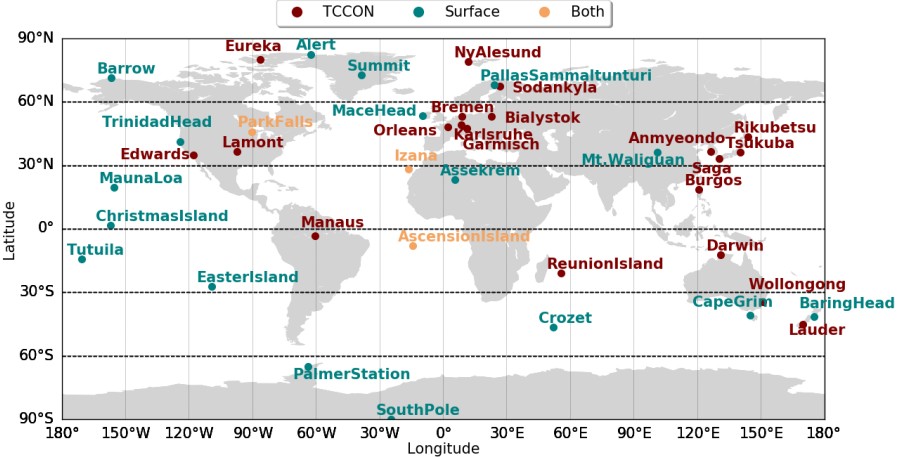

**Figure 11.** Locations of the flask surface sites from NOAA GGGRN (turquoise, Dlugokencky et al. (2020b, a); Petron et al. (2020)) along with sites that measure column-averaged dry air mole fractions as part of TCCON (red, https://tccondata.org/) and sites that are both part of TCCON and NOAA GGGRN (orange). For site details, see Table A1.

profile comparison we use aircraft measurements from the ATom campaigns (Wofsy et al., 2018). For $CO_2$ and CO we use the merged ATom data product collected from the NOAA-Picarro and Harvard Quantum Cascade Laser System instrument while

for $CH_4$ we use the measurements collected with the NOAA-Picarro instrument only.

We use column measurements from TCCON as the main data product to highlight the differences between the uncoupled and coupled simulations. Both the CO and $CO_2$ chemical sources are produced throughout the column, hence relative to surface measurements these measurements are more representative of the impact of chemical production on the total amounts of the gases. To compare the total CO and $CO_2$ model output with the column averaged measurements, we convert the modelled

mole fractions to column averaged dry-air mole fractions ($X_{gas}$) by dividing the vertical column of the gas of interest ($\Omega_{gas}$) with the total dry air column ($\Omega_{O_2}$), based on the method described by Wunch et al. (2010):

$$X_{gas} = 0.2095 \frac{\Omega_{gas}}{\Omega_{O_2}} \tag{10}$$

and smoothed according to Eq. (11) (Deutscher et al., 2014):

$$c_s = c_a + h^T a^T (x_m - x_a) \tag{11}$$

where $c_s$ represents the smoothed column model dry-air mole fraction, $c_a$ is the TCCON a priori column dry-air mole fraction, $h^T$ represents the vertical column summation, $a^T$ is the the Fourier-Transform Spectrometer averaging kernel, and $x_m$ and $x_a$ are the model and a priori dry-air mole fraction profiles.





The modelled vertical profiles are saved at a daily temporal resolution and extracted for the closest grid box to each TCCON station. For the comparison with surface measurements, we extract the grid points at the lowest level in the model. For compari-

son with aircraft measurements, model outputs are saved for grid boxes corresponding to the measured time, latitude, longitude and level along the plane flight track. Both the aircraft measurements and modelled output are averaged to the model temporal (20 min) and spatial ($2° \times 2.5°$) resolution to calculate one average value for each unique grid-box-time-step combination.

The 10 year spinup that is performed prior to the $CO_2$ and $CH_4$ simulations led to an offset between the simulated values and measurements. Due to the increasing trend of $CO_2$ and $CH_4$ in the atmosphere, each spinup year (repeating year 2005) adds

the yearly growth rate of 2005 to the modelled $CO_2$ and $CH_4$ values, leading to globally higher simulated values relative to the measurement. The global modelled growth of $CO_2$ and $CH_4$ in 2005 at the surface is 1.41 ppm and 0.96 ppb, respectively. In addition, for $CH_4$ the initial fields prior to the spinup are based on year 2010, introducing an additional offset. We quantify the overall offset by calculating the difference between the modelled $CO_2$ and $CH_4$ values used to initialize our uncoupled and coupled simulations (1st of January 2005) to measurements at different baseline NOAA GGGRN sites (Barrow, Mauna Loa,

American Samoa (Tutuila) and South Pole, average value for January 2005). The resulting offset is 14 ppm for $CO_2$ and 45.8 ppb for $CH_4$. We subtract this offset from the modelled values when comparing to surface, column and aircraft measurements.

Both the column and surface measurements are impacted by data gaps. To minimize the impact of the noncontinuous measurements and inconsistent measurement time periods on the analysis, we use a consistent time period (2010–2017) when analyzing the measurement-model differences. We find the fewest data gaps during this time period; however, a few sites are

still subject to missing measurements (column: Ny Alesund, Rikubetsu, Edwards, Anmyeondo, Saga, Ascension Island, Reunion; surface: Trinidad, Easter Island, Christmas Island). Due to short timeseries at the Manaus and Burgos TCCON sites, we exclude them from the plots representing the measurement-model differences in the next section (Sect. 6.1); however, the full timeseries at all sites can be found in Fig. S16–S21.

## 6.1 Comparison with column measurements

Figure 12 shows the differences between the modelled values (uncoupled, coupled and *coupled-origOH*) and measurements at different TCCON sites for $CH_4$ (a–e), CO (f–j) and $CO_2$ (k–o) plotted against the latitude of each site. We also show the normalized mean bias between the modelled and measured values on each plot. Mid-latitude European sites (Białystok, Bremen, Karlsruhe, Orléans and Garmisch, grouped into Other EU sites) show similar results, hence we only present their mean value. The timeseries comparison of the total CO, $CO_{CH_4}$, $CO_2$, $CO_{2CO}$ and $CH_4$ mole fractions for each site can be

found in Fig. S16–S18.

On a yearly scale, for all seasons the $CH_4$ values from the coupled simulation show noticeably better agreement with the measurements (Fig. 12a–e, indigo line) relative to the *coupled-origOH*. The *coupled-origOH* results show a positive bias with overestimated $CH_4$ values for all sites; however, using globally more abundant OH fields (v9-01-03) in the coupled simulation resolve this large bias. Although the coupled simulation shows better results, updating the OH fields does not resolve all the

differences, and a few NH mid-latitude sites (Anmyeondo, Tsukuba, Saga) still suggest a significant bias between the modelled and measured values, pointing to either underestimated $CH_4$ sources or overestimated sink fields (e.g., OH). Differences in



**Figure 12.** Column-averaged mole fraction model-measurement differences (uncoupled (red), coupled (indigo) and *coupled-origOH* (turquoise)) for CH$_4$ (a–e), CO (f–j) and CO$_2$ (k–o), as a function of latitude, averaged for 2010–2017 with annual values (a, f, k) and for different seasons: Dec–Jan–Feb (b, g, l) , March–April–May (c, h, m), June–July–Aug (d, i, n), Sept–Oct–Nov (e, j, o) . The numbers inset represent the Normalized Mean Bias (NMB). For CO we also show the NMB based on the unscaled CO values (shown in the parentheses).

the modelled-measured values throughout the seasons highlight potential contributors to the observed biases. For most SH (except Darwin during DJF) and NH mid-latitude sites (except Izana during DJF, MAM and JJA and Park Falls during JJA)



the coupled simulated values are lower relative to the measurements, with a smaller bias during SH autumn/winter (MAM,
JJA). In the NH polar regions we find a positive bias during NH summer (JJA) and negative bias during the other seasons.
During the growing season, $CH_4$ permafrost emissions are released in the Arctic regions (Laurion et al., 2010); however, this
emission is not included in the simulation. Adding this emission would make our simulated $CH_4$ values in the Arctic regions
even higher during summer, suggesting that either the bias in the polar regions is driven by biases in the sink fields throughout
the atmosphere or the missing permafrost signal is too weak to be detected in column measurements.

We find the largest CO model-measurement bias for the same NH mid-latitude sites that display a large $CH_4$ bias (Fig.
12f–j). The smallest bias is present at sites closest to the South Pole with increasing negative bias (i.e., underestimation of
the measured values) towards the NH; however, in the NH the biases show a smaller latitudinal dependence than in the SH,
presumably due to the larger differences in the CO sources between regions/sites. In both hemispheres and all seasons (except
SH mid-latitude regions, Lauder and Wollongong) the coupled results show the best agreement with measurements as a result
of stronger CO production from $CH_4$. In the SH we find a stronger negative bias during spring (SON) periods, while the
seasonal dependence in the NH is more variable. The largest bias relative to the measurements in the NH and SH is from the
*coupled-origOH* and uncoupled simulation, respectively. Differences in the CO values are driven by differences in the $CH_4$
loss calculation. We find that the stronger $CH_4$ loss in the coupled simulation leads to better agreement with measurements,
suggesting that this term was underestimated in the other simulations. Previous studies showed that CO values in the SH are
dominated by $CH_4$ and NMVOC oxidation (Zeng et al., 2015; Té et al., 2016; Fisher et al., 2017), hence underestimated
secondary CO production values are potentially the origin of the remaining model-measurement bias in the SH. Moreover,
the largest bias in the SH is observed during spring (SON), suggesting an additional underestimated biomass burning source,
since this period aligns with the burning season in the SH. Note, due to potential errors in the TCCON column CO scaling
factors we also compare our modelled CO with the unscaled column CO values that are higher by $\approx 7\%$. For all sites, we
obtain the unscaled values by multiplying the column CO by 1.0672. The unscaled values will further increase the negative
model-measurement bias.

For most sites and seasons, the simulated $CO_2$ values are higher than the measurements (Fig. 12k–o); however, a few sites
in the NH tropical and polar, as well SH mid-latitude, regions show lower $CO_2$ values in the simulation (Anmyeondo, Saga,
Eureka and Lauder based on the annual values). Not taking into account these four sites, we find a latitudinal dependence of
the bias, increasing from the SH polar regions towards the North Pole. In the SH, a large bias is seen during autumn/winter
periods (MAM, JJA) in tropical regions, potentially driven by an underestimated terrestrial $CO_2$ sink in the tropics. This bias
is potentially even stronger in the NH tropical regions; however, we cannot confirm this due to a lack of TCCON sites in
this region. The NH mid-latitude regions show a strong bias during all seasons, with both overestimated and underestimated
$CO_2$ values. Both the coupled and *coupled-origOH* simulations show better agreement for most sites while the uncoupled
simulation results show the largest bias. The main difference between the uncoupled vs coupled $CO_2$ values is the weaker
$CO_2$ chemical production in the coupled simulations, suggesting that this source term might have been overestimated in the
uncoupled simulation.





Overall, our coupled simulation show better agreement with column measurements (smallest model-measurement bias) than the original uncoupled simulation. The coupling improves the model-measurement offset; however, the distribution of

the differences between sites are consistent between different simulation versions. Further bias reductions would come from reducing uncertainties in other fluxes and transport. The inclusion of an OH-feedback between species would also impact the model-measurement bias, especially during enhanced localized emission events (i.e., fires). As an example, strong CO emissions would lead to depleted OH values, resulting in weaker oxidation of $CH_4$ and production of $P(CO)_{CH_4}$. This feedback is not captured in either of the simulations since the OH fields are fixed and separated between species.

**6.2  Comparison with surface measurements**

We complement the column measurements with surface measurements to highlight the impact of the chemical sources at the surface relative to the column. Figure 13 shows the differences between the measurements versus the uncoupled, coupled and *coupled-origOH* simulations at different surface sites for $CH_4$ (a–e), CO (f–j) and $CO_2$ (k–o) plotted against the latitude of each site. The timeseries comparison for each site can be found in Fig. S19–S21. Note, relative to the column results the surface

comparison is more strongly impacted by the coarse model resolution ($2°x2.5°$). The measured and modelled column values are more representative of regional and larger scale processes so the impact of the model resolution is weaker.

In some regions the surface model-measurement comparison show differences relative to the column comparisons, highlighting the importance of using both surface and column measurements for validation, and for identifying potential processes that drive the observed differences. Similar to the column comparison, for $CH_4$ the surface values from the coupled simula-

tion show better agreement with measurements; however, the surface $CH_4$ values are consistently higher (positive bias) in the simulations, while in the column results we find lower simulated $CH_4$ values (negative bias). This difference is potentially due to the mis-representation of $CH_4$ in the stratosphere that only impacts the column data or to stratospheric biases due to insufficient spin up period.

For CO, the NH biases are similar to the column comparison, with the best agreement for the coupled results. For the SH

the bias between the surface and column comparison differ between simulations. In contrast to the column results where, on average, the coupled shows the best agreement, at the surface we find the best agreement with the *coupled-origOH* result (followed by coupled) in SH tropical regions, and uncoupled results (followed by coupled) in the remaining SH regions. Moreover, the column CO values are lower than the measurements; however, at surface level we see an overestimation of the CO values for some SH sites, although a number of these sites is in a region where we lack column measurements. The

overestimated values in the SH might be partially due the weaker vertical mixing in the model leading to buildup of CO in the planetary boundary layer.

For $CO_2$, we find similar results between the surface and column, with overall a larger range of biases in the surface data. The strong negative bias seen at some NH sites in the column comparison is not present in the surface comparison. This difference is potentially due to the diel cycle, which is less pronounced in the column data, coupled with an incorrect flux distribution.

In the SH, we see a negative bias for sites between 45–90° S, a region where we lack column measurements. The differences



**Figure 13.** Surface mole fraction model-measurement differences (uncoupled (red), coupled (indigo) and *coupled-origOH* (turquoise)) for $CH_4$ (a–e), CO (f–j) and $CO_2$ (k–o), as a function of latitude, averaged for 2010–2017 with annual values (a, f, k) and for different seasons: Dec–Jan–Feb (b, g, l) , March–April–May (c, h, m), June–July–Aug (d, i, n), Sept–Oct–Nov (e, j, o). The numbers inset represent the Normalized Mean Bias (NMB).

in the polar regions are potentially impacted by additional $CO_2$ exchange from air–sea ice interaction, a process that is not included in the simulation, and still subject to large uncertainties (Søgaard et al., 2013).





Overall, our simulation updates lead to better representation of observed $CO_2$, $CH_4$ and CO for most regions and seasons through the updated calculations of $P(CO)_{CH_4}$ and $P(CO_2)$ and through using updated OH fields. However, biases relative

to the measurements still remain, especially in the NH. For all three gases, we find the smallest biases in the SH, with an increasing trend towards the North Pole. Different biases in the surface and column comparison suggest that potential biases in vertical transport should also be explored.

### 6.3 Comparison with aircraft measurements

We further compare the simulations with aircraft measurements collected as part of ATom (campaign 1: July–August 2016,

2: January–February 2017, 3: September–October 2017 and 4: April–May 2018) during 2016–2018. Figure 14 shows the differences between the modelled and measured $CH_4$, CO and $CO_2$ values during the four campaigns as a function of latitude and pressure. The spatial distribution of the differences between the modelled and measured values is shown in Fig. S22.

The latitudinal change of the aircraft model-measurement differences, for all three gases and simulations follows the pattern seen in the column data. For $CH_4$ and CO (on average) the coupled simulation shows the closest values to measurements, with

lower modelled values. The negative CO bias is present during all seasons and for latitudinal bands except during ATom-2 (SH summer) and ATom-4 (SH fall) south of 50–60°. In the column data we do not have sites south of 45° and in tropical regions in the NH and cannot identify these biases. For $CO_2$ we find a negative bias in the SH and positive bias in the NH with similar magnitude. Both the coupled and *coupled-origOH* simulations result in almost the same $CO_2$ values, while the uncoupled results show the same distribution of the biases but with a consistent offset with overall better estimates in the SH

and larger positive bias (relative to the measurements) in the NH.

The altitudinal distribution of the model-measurement biases is similar between simulations. For $CH_4$ there is no significant change in the vertical distribution of the biases. Using different OH fields as before impacts the overall model-measurement bias, switching from a positive to a negative bias but it has a small impact on the vertical distribution of the biases. For CO, the coupled simulation shows better estimates during all campaigns except ATom-4 during June–July when the model-

measurement bias is the smallest in the *coupled-origOH* simulation. Overall, the modelled CO values underestimates the measurements during all four campaigns/seasons and vertical levels. Differences between the three simulations reduce approaching higher altitudes in the model. The vertical distribution of the $CO_2$ biases only differs between simulations in the offsets.

Figure 15 shows the altitude versus latitude cross-sections of the $CO_2$ and CO chemical tracers along the ATom flight tracks.

The cross-sections are constructed from linear interpolation between modelled values at the aircraft measurement locations after averaging the modelled data as described in Sect. 6.

The $CO_{CH_4}$ values in the coupled simulation reflect the seasonal pattern in the production fields shown in Fig. 10. The mole fractions are highest in the NH during the NH summer (ATom-1) followed by NH fall (ATom-3). During NH winter (ATom-2), we see a shift of the production from the Northern to the Southern Hemisphere while during NH spring (ATom-4) we find high

mole fraction values in the tropical regions. The uncoupled simulation shows a similar distribution but with lower values, while the *coupled-origOH* shows higher values in the SH and lower values in the NH during all campaigns except ATom-4.





**Figure 14.** Aircraft model-measurement (uncoupled (red), coupled (indigo) and *coupled-origOH* (turquoise)) $CH_4$, CO and $CO_2$ differences, shown as their latitudinal ($CH_4$: a, g, m , s, CO: b, h, n, t, $CO_2$: c, i, o, u) and altitudinal distribution ($CH_4$: d, j, p, v, CO: e, k, q, w, $CO_2$: f, l, r, x) during the four ATom campaigns in June–July 2016 (a–f), Dec–Jan 2017 (g–l), Aug–Sep 2017 (m–r) and Mar–Apr 2018 (s–x). Horizontal lines show standard deviation within each bin. The data is averaged into 10° latitudinal and 50 mb pressure bins. The numbers inset represent the Normalized Mean Bias (NMB).

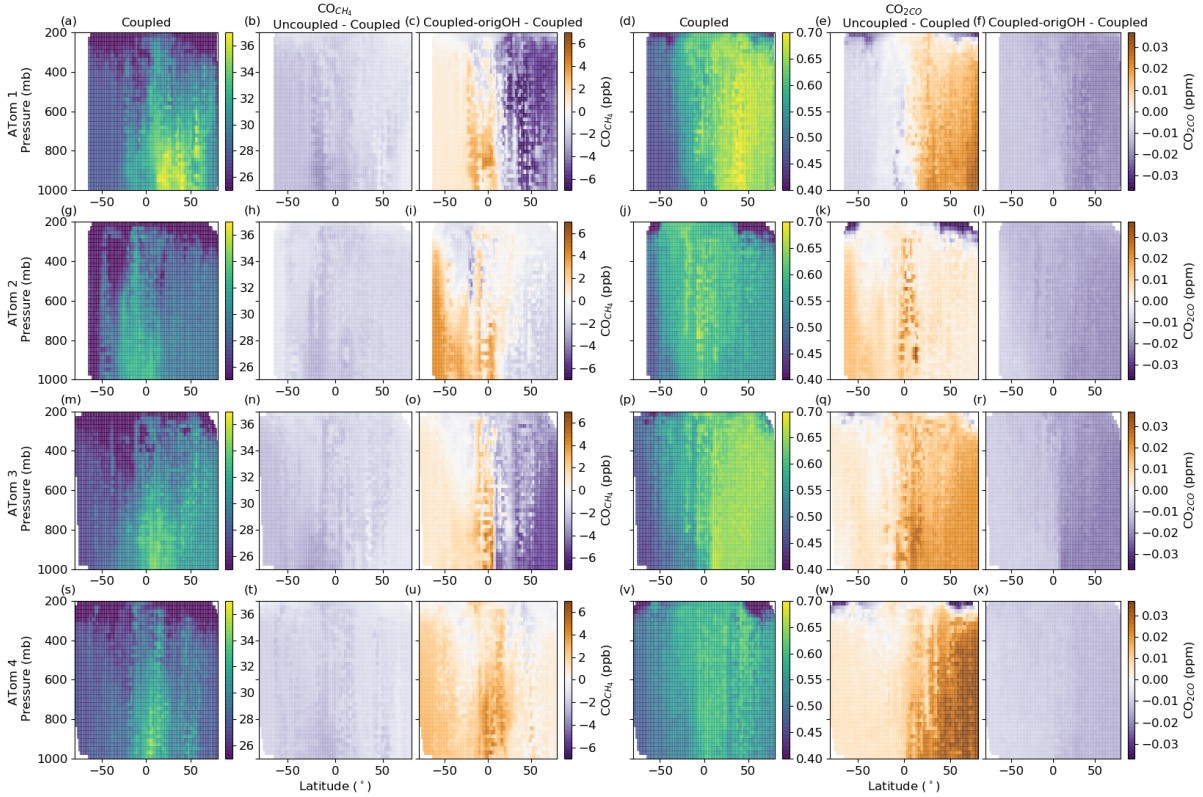

**Figure 15.** Altitude versus latitude cross-sections of chemically produced CO ($CO_{CH_4}$, a–c, g–i, m–o, s–u) and $CO_2$ ($CO_{2CO}$, d–f, j–l, p–r, v–x) mole fractions from the coupled simulation along with the uncoupled and *coupled-origOH* differences relative to the coupled simulation during the 4 ATom campaigns.

The modelled $CO_{2CO}$ values are more mixed throughout the atmosphere. As with $CO_{CH_4}$, we observe the highest $CO_{2CO}$ values in the NH during NH summer followed by NH fall. In the SH, $CO_{2CO}$ is highest during SH summer (ATom-2) but not as high as the NH summer values in the NH. The *coupled-origOH* simulation shows a similar distribution but with lower values, while the uncoupled simulation points to overall higher values globally except during June–July (ATom-1) when the mole fractions are lower in the SH. As shown in Fig. 14, the differences in the chemical fields between simulations are also reflected in the total values of CO and $CO_2$.

# 7   Conclusions

We developed a coupled carbon greenhouse gas simulation in the GEOS-Chem chemical transport model that combines $CO_2$, $CH_4$ and CO, through their chemical dependence. The coupling between the three gases is from the chemical production of CO from $CH_4$ loss fields ($P(CO)_{CH_4}$) and the chemical production of $CO_2$ from the oxidation of CO ($P(CO_2)$). In the





previous uncoupled version of these simulations, the chemical productions were handled offline based on monthly archived fields for specific years from older model versions, while in our coupled simulation we calculate $P(CO)_{CH_4}$ and $P(CO_2)$ at every model timestep.

Along with the uncoupled versions of $CO_2$, $CH_4$ and CO simulations, we run two versions of the coupled simulation using (i) updated and consistent OH fields (v9-01-03) for the calculation of $CH_4$ and CO loss (coupled) and (ii) using disconnected OH fields (v5-07-08) for the calculation of $CH_4$ loss to match the OH used in the original uncoupled simulation (*coupled-origOH*). We compare the uncoupled and coupled results to identify the impact of the online calculation of the chemical terms, while the *coupled-origOH* results are used to identify the impact of inconsistent OH fields between species.

Our budget estimates from the coupled simulation agree with known literature values. For the 2006–2017 time period our coupled results show an increase in $P(CO)_{CH_4}$ with time due to the interannual variability of $CH_4$ loss and a dependence on climate anomalies (i.e., El Niño Southern Oscillation). We find differences between the coupled and uncoupled simulations ranging from 29–61 Tg CO $yr^{-1}$ (1.2–2.6 % of the total CO source in the coupled version). The stronger production in the coupled simulation is a result of using updated and globally higher OH values for the calculation of $CH_4$ loss. Our $P(CO_2)$

from the coupled simulations are weaker than in the uncoupled simulation, with a 0.04–0.09 Pg C $yr^{-1}$ difference (0.3–0.7 % of the total $CO_2$ source in the coupled version) but with stronger production in tropical land regions.

     We find the choice of the OH fields between simulations and species has a significant impact on the differences in the chemical terms and resulting mole fractions. Globally, the $P(CO)_{CH_4}$ fields are significantly different between the coupled and *coupled-origOH* simulations due to different OH fields, leading to large differences in the modelled mole fractions of the

chemical tracers. However, the choice of OH field when calculating the $CH_4$ loss has a negligible impact on $P(CO_2)$ since it only represents an indirect and minor contribution when calculating $P(CO_2)$

     Our $CH_4$, CO and $CO_2$ values from the coupled simulation overall show better agreement with all three measurement products we use for the validation (TCCON column measurements, NOAA GGGRN surface measurements and ATom aircraft data). The exception are tropical surface sites in the SH and sites between 40–90° S, where the *coupled-origOH* and uncoupled

CO values show better agreement, respectively. The uncoupled simulation also shows better agreement for surface $CO_2$ in the 45–90° S region, as well in the SH when comparing to aircraft data. These findings point to further biases that were previously masked when using only the uncoupled simulation through the limited simulation of the chemical terms.

     Based on the model-measurement biases we find that the default v5-07-08 OH fields were incorrect when calculating the $CH_4$ loss. Increasing the OH concentrations with a seasonality of maximum surface value in May improves both our $CH_4$ and

CO modelled values. For CO, the biases in the SH can partially be explained by underestimated biomass burning emissions, especially during the dry season, and underestimated secondary CO production ($CH_4$ and NMVOC oxidation) values. Our coupled simulation suggests that the chemical production of $CO_2$ in the Amazon was significantly underestimated in previous $P(CO_2)$ studies (Nassar et al., 2010). We overall find stronger $P(CO_2)$ above tropical land regions. South America, Central Africa and northern Australia are characterized by strong biomass burning and our coupled model simulates the $P(CO_2)$

in these regions during the fire season to be stronger than in previous fields, while the stronger $P(CO_2)$ in regions such as East Asia and North America points to enhanced anthropogenic CO emissions. For $CO_2$ inclusion of the missing exchange





from air–sea ice interaction can potentially contribute to better modelled values in the polar regions. Differences in the model-measurement biases between the column and surface data also point to the mis-representation of $CH_4$ in the stratosphere and biases in the vertical mixing that impacts all three gases. Excluding the OH feedback can also lead to persistent biases in the

modelled values, especially in regions where chemical production/loss is enhanced.

The newly developed coupled simulation enables future investigations of the co-variations of $CO_2$, $CH_4$ and CO, as well as their interannual variability, that will provide better understanding of their interactions. We have shown that coupling the three gases result in more accurate modelled values and improves our ability to identify source and sink fields that are over- or underestimated in the model. Although our updates include better model-measurement agreement for all three gases, biases

still remain, highlighting the importance of further improvement of these simulations. These differences are heavily influenced by the existing uncertainties in variety of carbon gas sources and sinks (Dlugokencky et al., 2011; Bukosa et al., 2019; Bastos et al., 2020). The new coupled simulation paves the way for future improvements, including inclusion of a $CH_4$–OH–CO feedback and implementation into the GEOS-Chem Adjoint used for inverse modelling, that will further improve our ability to constrain the fluxes of the carbon gases. With updates such as this simulation we will be able to better highlight and identify

the origin of the model-measurement differences and constrain the sources, sinks and budgets of $CO_2$, $CH_4$ and CO, crucial for future climate projections and mitigation policies.

**Appendix A: Appendix A**



**Table A1.** Column and surface stations used for the coupled simulation validation. Sites are ordered based on latitude, from highest to lowest.

| Station | Latitude | Longitude | Elevation (m) |
|---|---|---|---|
| ***TCCON sites*** | | | |
| Eureka[a] | 80.05° N | 86.42° W | 610 |
| Ny Alesund[b] | 78.90° N | 11.89° E | 20 |
| Sodankyla[c] | 67.37° N | 26.63° E | 188 |
| Białystok[d] | 53.23° N | 23.02° E | 180 |
| Bremen[e] | 53.10° N | 8.85° E | 27 |
| Karlsruhe[f] | 49.10° N | 8.43° E | 116 |
| Orléans[g] | 47.97° N | 2.11° E | 130 |
| Garmisch[h] | 47.48° N | 11.06° E | 740 |
| Rikubetsu[i] | 43.46° N | 143.77° E | 380 |
| Lamont[j] | 36.60° N | 97.49° W | 320 |
| Anmyeondo[k] | 36.54° N | 126.33° E | 30 |
| Tsukuba[l] | 36.05° N | 140.12° E | 30 |
| Edwards[m] | 34.96° N | 117.88° W | 699 |
| Saga[n] | 33.24° N | 130.29° E | 7 |
| Burgos[o] | 18.53° N | 120.62° E | 35 |
| Manaus[p] | 3.21° S | 60.60° W | 50 |
| Darwin[q] | 12.43° S | 130.89° E | 30 |
| Reunion Island[r] | 20.90° S | 55.48° E | 87 |
| Wollongong[s] | 34.41° S | 150.88° E | 30 |
| Lauder[t] | 45.04° S | 169.68° E | 370 |
| ***Both TCCON sites and surface[x,y,z]*** | | | |
| Park Falls[u] | 45.94° N | 90.27° W | 440 |
| Izana[v] | 28.30° N | 16.50° W | 2370 |
| Ascension Island[w] | 7.91° S | 14.33° W | 10 |
| ***Surface sites[x,y,z]*** | | | |
| Alert | 82.45° N | 62.51° W | 185 |
| Summit | 72.50° N | 38.42° W | 3209 |
| Barrow | 71.32° N | 156.61° W | 11 |
| Pallas Sammaltunturi | 67.97° N | 24.12° E | 565 |
| Mace Head | 53.33° N | 9.89° W | 5 |
| Trinidad Head | 41.06° N | 124.15° W | 107 |
| Mt. Waliguan | 36.29° N | 100.89° E | 3810 |
| Assekrem | 23.26° N | 5.63° E | 2710 |
| Mauna Loa | 19.53° N | 155.58° W | 3397 |
| Christmas Island | 1.70° N | 157.15° W | 0 |
| Tutuila | 14.25° S | 170.56° W | 42 |
| Easter Island | 27.16° S | 109.43° W | 47 |
| Cape Grim | 40.67° S | 144.69° E | 94 |
| Baring Head | 41.41° S | 174.87° E | 85 |
| Crozet | 46.43° S | 51.84° E | 197 |
| Palmer Station | 64.77° S | 64.05° W | 10 |
| South Pole | 89.98° S | 24.80° W | 2810 |

[a] Strong et al. (2019) [b] Notholt et al. (2019b) [c] Kivi et al. (2014) [d] Deutscher et al. (2015) [e] Notholt et al. (2019a) [f] Hase et al. (2015) [g] Warneke et al. (2014) [h] Sussmann and Rettinger (2018) [i] Morino et al. (2018c) [j] Wennberg et al. (2016) [k] Goo et al. (2014) [l] Morino et al. (2018a) [m] Iraci et al. (2016) [n] Kawakami et al. (2014) [o] Morino et al. (2018b) [p] Dubey et al. (2014) [q] Griffith et al. (2014a) [r] De Mazière et al. (2017) [s] Griffith et al. (2014b) [t] Sherlock et al. (2014) [u] Wennberg et al. (2017) [v] Blumenstock et al. (2017) [w] Feist et al. (2014) [x] $CO_2$: Dlugokencky et al. (2020b) [y] $CH_4$: Dlugokencky et al. (2020a) [z] CO: Petron et al. (2020)



*Code and data availability.* GEOS-Chem in an open-source model, the original v12.1.1 model is publicly available at https://doi.org/10.5281/zenodo.2249246. The exact version of the model used to produce the results used in this paper is archived on Zenodo (https://doi.org/10.5281/zenodo.5077000), as are input data and scripts to run the model and produce the plots for all the simulations presented in this paper. The coupled simulation will also be implemented in an upcoming newer version of GEOS-Chem. Additional GEOS-Chem model output is available from the authors upon request. TCCON data is publicly available at https://tccondata.org/. NOAA GGGRN surface data is publicly available at https://www.esrl.noaa.gov/gmd/dv/data/. ATom data is publicly available at https://doi.org/10.3334/ORNLDAAC/1581.

*Author contributions.* BB developed the code for the coupled the GEOS-Chem simulation, ran the model, performed the analysis and led the writing of the paper under the supervision and guidance of NMD and JAF. DBAJ helped formulating the method for the coupled simulation. All authors contributed to editing and revising the paper.

*Competing interests.* The authors declare that they have no conflict of interest.

*Acknowledgements.* We thank Clare Paton-Walsh and Sara E. Mikaloff Fletcher for valuable discussions about this paper. We thank the TCCON PIs and funding agencies for the TCCON measurements. We thank the NOAA ESRL Global Monitoring Laboratory, Boulder, Colorado, USA, for providing the different surface site data. We acknowledge the ATom Team for providing the aircraft data from all four campaigns. This research has been supported by the Australian Research Council (grant nos. DE140100178, DP160101598 and FT180100327), Discovery Early Career Researcher (DECRA) University Postgraduate Award from the University of Wollongong, assistance of resources provided at the NCI National Facility systems at the Australian National University through the National Computational Merit Allocation Scheme supported by the Australian government (grant no. m19) and funding from New Zealand's Ministry of Business, Innovation and Employment through contract number C01X1817.



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
