# Peer review of "An improved carbon greenhouse gas simulation in GEOS-Chem version 12.1.1"

_Geoscientific Model Development, 2021_

## Author Comment (AC1)

**In the text below we have included all the referee comments in black, followed by our response in red.**

**Anonymous Referee #1**

A new greenhouse gas simulation in GEOS-Chem is presented, which couples CO2, CH4 and CO. The paper presents a detailed evaluation of the new system and compares the model simulations to observations.

It seems likely that this new framework will be useful for inverse modelling studies in the future. The analysis is very thorough, although perhaps some aspects could be cut or moved to the Supplement to help streamline the text (which is quite long). However, I have some concerns about the experiment design and interpretation. I feel that the paper would suitable for publication in GMD after these concerns have been addressed.

General comments

Experiment design: The novel component of the paper is the description of the coupled CO2-CO-CH4 system. However, I found the discussion of the comparisons of the coupled and uncoupled systems very hard to follow in places. I think this is primarily because the various simulations were run with different OH fields and model resolutions (and with/without an OH diurnal cycle, although this doesn't appear to have much impact). To partially address this, an additional OH run is carried out to test the sensitivity to OH, introducing another model into the mix. It seems that the reason for these inconsistencies is historical, although I wasn't entirely sure from the text. It seems to me that this paper could be dramatically improved, streamlined and clarified by carrying out two sets of model runs, using the same OH field, the same resolution, one run coupled and one uncoupled. Given that it's relatively cheap to run these simulations, I'm not sure why this wasn't done. Can the authors explain? If it's to maintain traceability back to earlier GEOS-Chem simulations, I'd argue that this should be a secondary consideration, compared to showing the improvement due to the new simulation setup.

**We thank the reviewer for the comments and discussion. This comment (and those from the other reviewers) has highlighted to us that perhaps the manuscript did not clearly describe the aim of our work, and this has led to some confusion about the intent of our work and therefore the experimental design. We have substantially revised the manuscript to better clarify both of these points. Here, we summarise our rationale along with changes we have made in the manuscript.**

**There was a strong reason for the experimental setup presented in the paper. The aim of the work was to compare the *existing* GEOS-Chem CH₄, CO, and CO₂ simulations that are currently being used by the community to the improved, coupled simulation we have developed. The existing simulations have multiple inconsistencies – in addition to the chemistry being uncoupled, they rely on inputs created at different points in time from different versions of the full chemistry model (including different OH, but also other differences in the chemical schemes). Our goals were to 1) present the problems associated with the existing uncoupled simulations and the way they are currently being used in the GEOS-Chem community and 2) present the improvements introduced by the coupled simulation. We now clarify this explicitly in the revised manuscript.**

**What we wanted to demonstrate is that if one uses the uncoupled GEOS-Chem CH₄, CO, and CO₂ simulations out-of-the-box, they are not consistent, introducing unnecessary errors when using these simultaneously for a single dataset, inversion, etc. If we were to change to consistent OH fields between the uncoupled (i.e., running consistent full chemistry simulations to produce the**

offline chemical fields) and coupled simulation, this would disguise the key limitations and biases in the chemical production/loss in the uncoupled simulations.

Another factor that we did not make entirely clear is that when we referred to the OH being from a particular model version, what this really means is that the chemical production fields were calculated using that model version. The OH distributions are a major contributor to the difference in chemical production between versions, but there will also be additional differences embedded in the chemical production fields (e.g., different emissions). To summarize our experimental setup: the uncoupled simulations use archived chemical production fields created in separate full-chemistry simulations based on what was available at the time each individual offline simulation was most recently updated (CO (v9) and $CO_2$ (v8)). Recreating all the chemical production fields with the same OH distribution would require re-running the full chemistry simulations for the entire period, which would be significantly more computationally expensive than the offline simulations. Again this would be of limited value, as we would effectively be creating a false narrative: comparing our new coupled simulation to individual $CH_4$, CO, and $CO_2$ simulations that do not exist and are not being used by the community.

In the new coupled simulation, we are calculating (online) the chemical production of CO (from loss of $CH_4$) and $CO_2$ (from loss of CO) using consistent (v9) OH fields and comparing these to the out-of-the-box uncoupled CO and $CO_2$ simulations to demonstrate the inconsistencies and biases if the GEOS-Chem user runs the default out-of-the-box uncoupled simulations. One thing that comparison does not highlight is inconsistencies between the out-of-the-box $CH_4$ and CO simulations. The out-of-the-box uncoupled $CH_4$ simulation uses v5 OH, while the out-of-the-box uncoupled CO simulations uses $P(CO)_{CH4}$ based on a full chemistry simulation with v9 OH. This means that in the uncoupled simulations, $L(CH_4)$ and $P(CO)_{CH4}$ are *not* equivalent (although, in reality, they should be). The reason we added the additional (orig-OH) coupled simulation was to demonstrate the impact of this discrepancy, by determining what the coupled simulation would look like if we calculated $P(CO)_{CH4}$ (and follow-on parameters) from the same $L(CH_4)$ as in the out-of-the-box $CH_4$ simulation. We understand that this is a fairly subtle point, and that its inclusion alongside the coupled and uncoupled simulations may have hampered the readability of the manuscript.

To improve the paper, we have made a number of changes in the revised manuscript. In addition to clarifying the above points throughout, we have also made the following substantial changes:

1. We focus the analysis on comparison of the uncoupled and coupled simulations only, removing the orig-OH simulation from the main comparison discussion and figures. Instead, we have added a short subsection with discussion about the orig-OH simulation using a 1-year simulation to more clearly make the point outlined above. By focusing on the coupled and uncoupled simulation results, we now clearly show the value of our improved simulation relative to the out-of-the-box simulations.
2. To simplify the analysis, we now focus the results on CO and $CO_2$ (and their chemical production terms) and remove parts of the $CH_4$ analysis.

Comparison to observations: The conclusions state that the new model improves the fit to the observations (L488 – 490). However, I don't think that this can be concluded. Since we do not know the "true" flux magnitude for these gases, we can't be sure that the coupled simulations are really improving the fit to the data, or just compensating for some bias(es) in the flux fields. For example, for methane, it is stated (L446) "The coupled-origOH results show a positive bias with overestimated CH4 values for all sites; however, using globally more abundant OH fields (v9-01-03) in the coupled simulation resolve this large bias". However, it could well be that the new fields are simply compensating for some bias in the (highly uncertain) emission field.

**We agree with the reviewer. In the revised manuscript, we rephrase and correct these statements and highlight that reduced bias in the coupled simulation could be compensating for biases in the emissions.**

Specific comments:

L107 (Eq. 2): I don't think this equation works. For closure, I think there also needs to be a term representing the net flux from/to the troposphere.
**We thank the reviewer for alerting us to the fact that we did not adequately describe this equation (or indeed the other equivalent equations) in the manuscript. We now explicitly state in the revised manuscript that these equations only describe the changes in the emission, deposition, production and loss terms that occur within each grid box and that advective transport fluxes between grid boxes (including between the troposphere and the stratosphere) are in addition to the terms described in each equation.**

L113: [OH] has already been defined.
**[OH] definition removed.**

L129 (Eq. 7): Again, need flux to/from the troposphere.
**See above; we now clarify this for all relevant equations.**

Figure 3: Is this figure relevant?
**Figure 3 highlights well the spatial differences in OH between the coupled and uncoupled simulation. Differences in the OH spatial distribution have a direct impact on the spatial distribution and analysis/understanding of the CO and $CO_2$ chemical fields. However, due to the reduction of the orig-OH simulation analysis we move both Figure 2 and 3 to the Supplement.**

L226 – 227: Some of this inter-annual variability is present in the uncoupled simulation. Is the change really so marked in the coupled version?
**We certainly observe stronger inter-annual variability in the coupled simulation, as well as changes in the chemical fields that are not present in our uncoupled simulation. In response to comments from Reviewer #3, we have clarified this discussion to explain the source of the inter-annual variability in the uncoupled simulation.**

L231 and Figure 5 caption: The figure shows the CO production, not the "Changes of the CO production". The change can be inferred from the figure, but is not directly shown.
**We have removed 'changes' from both L231 and Figure 5 caption.**

Figure 4: Add an x-axis. Also, it's not clear why a bar chart is the best way to present this. How about a line graph?
**We have added the years as the x-axis (and axis label 'Years'). Figure 4 shows the global summary of the different region budgets presented in the Supplement (Fig S3-S6) that are difficult to show as a line graph. Hence for consistency we used the same plotting procedure (i.e., bar graph with the same colouring). We now state in the figure caption *"Regional distributions are shown in Figs. S3-S6 in the Supplement"* to better make the link between this figure and the ones in the Supplement.**

L244: "Hemispheric" seems preferable to "regional" to describe the table.
**'regional' changed to 'hemispheric' in both the text and Table 2 caption.**

Figure 7: Perhaps move this, and the discussion, to the Supplement. I'm not sure it adds too much.
**In the revised manuscript, both Figure 7 and the related discussion have been removed from the main text.**

L340: I think a comma would be preferable to a semi-colon between "seasons" and "but".
**Changed to a comma.**

L431: The offset in the modelled values will lead to a small difference in the CO production, etc., compared to the real atmosphere (i.e. CO production from CH4 will be over-estimated, since the model spin up leads to higher CH4). Is this effect important?
**We thank the reviewer for highlighting this. The higher initial $CH_4$ levels (due to the spin up) in the coupled simulation will indeed lead to stronger CO production after reaction with OH. This was an error in our original work that we have now corrected. To correct this, we re-ran the coupled and uncoupled $CO_2$ simulations with all the offsets (in both the $CH_4$ and $CO_2$ initial fields) corrected prior to the simulations (instead of the post-simulation correction we used previously). The figure below shows the original results (with the offset present in the chemical term) and the new corrected result. Using the corrected restart files leads to lower $P(CO)_{CH_4}$ values; however, the overall analysis in the paper remains the same, since the coupled results still show higher values relative to the uncoupled CO simulation and follow the inter-annual variability as in the original analysis. In the revised manuscript we have updated all the results to use the corrected simulation and we modified the description of the offset correction.**

[Figure]

L449: "resolves", rather than "resolve" (although, see general comment... I don't agree with this statement!)
**We have rephrased this statement, as described in our response in the general comment (comparison to observation).**

L464 / 465: Notwithstanding the issues around flux magnitudes, these differences seem very small compared to all the other uncertainties in the system. I think it'd also be fair to say that the model changes had a negligible impact on the comparison with the observations here.
**We have rephrased/removed this based on our response above to the general comment (comparison to observations).**

L488 – 491: I think these lines in particular (and many others throughout) need to be revised in light of my general comment regarding the potential impact of uncertain fluxes.
**As above, we have rephrased this statement.**

L593: I don't think you can say that the v5-07-08 fields are incorrect based on this analysis, given the emissions uncertainty.
**In this specific case, we have robust and strong evidence that the v5-07-08 fields are incorrect. The chemical scheme used in GEOS-Chem v5-07-08 (circa 2004) was much simpler than modern chemical schemes and is now extremely outdated. For example, in v5-07-08 the model**

**underestimated lightning NOx emissions, and thus ozone, in the northern extratropics. Further, there is clear evidence that the v5-07-08 OH fields are incorrect since they produce an unrealistic seasonal cycle in $CO_{CH4}$ that is corrected when using newer OH fields. We now explain this in the new dedicated sub-section on the orig-OH simulation.**

L607 – 609: As above, I think this conclusion needs to be removed.
**As above, we have rephrased this statement.**

Appendix A: Titled as "Appendix A: Appendix A"
**Fixed.**

---

## Author Comment (AC2)

**In the text below we have included all the referee comments in black, followed by our response in red.**

**Anonymous Referee #2**

Review of "An improved carbon greenhouse gas simulation in GEOS-Chem version 12.1.1" by Beata Bukosa et al.

This paper simulated CO2, and CO2 production by oxidation of CH4 and CO using 3 different types of OH fields. The topic of research is important for better estimation of CO2 sources and sinks on the Earth's surface. The distribution of OH is heavily concentrated over the tropical region, where CO2 would be added to the atmospheric CO2 due to oxidation of CO and CH4. If ignored this CO2 chemical production, biased source of CO2 is needed from the tropical land and ocean regions by inverse modelling. I very much liked the idea of this research, but unfortunately wasn't able to read through the whole manuscript due to poor execution of the research idea, in my opinion. Thus I cannot recommend publication of this work in Geoscientific Model Development in the present form or anything close to this. It is better to rerun the model and submit a newly prepared manuscript.

**We believe the reviewer has misunderstood both the aim of our research and the experimental design and execution. The reviewer's misunderstanding (and comments from the other reviewers) has highlighted to us that perhaps the manuscript did not clearly describe the aim of our work, and this has led to some confusion. We have substantially revised the manuscript to better clarify both the intent and the experimental design of our work. Here we summarise these points before addressing the reviewer's specific comments. (See also our response to Reviewer #1 for a more detailed description.)**

**The aim of our work was to compare the _existing_ GEOS-Chem $CH_4$, CO, and $CO_2$ simulations (which are uncoupled) that are currently being used by the community to a new version of the simulation that couples these species together based on their chemistry. The existing simulations have multiple inconsistencies – in addition to the chemistry being uncoupled, they rely on inputs created at different points in time from different versions of the full chemistry model (including different OH, but also other differences in the chemical schemes). Our goals were to 1) present the problems associated with the existing uncoupled simulations the way they are currently being used in the GEOS-Chem community and 2) present the improvements introduced by the coupled simulation. The intent was not to focus on the differences in OH, and we now clarify this in the revised manuscript.**

**We also note that the original (uncoupled) simulations do not ignore $CO_2$ chemical production as implied by the reviewer. Rather, the $CO_2$ production in the uncoupled $CO_2$ simulation is inconsistent with the CO chemical loss in the uncoupled CO simulation (and the same is true for CO production from $CH_4$). It is this inconsistency our work aims to rectify.**

**To improve the paper, we have made a number of changes in the revised manuscript. In addition to clarifying the above points throughout, we have also made the following substantial changes:**

1. **We focus the analysis on comparison of the uncoupled and coupled simulations only, removing the orig-OH simulation from the main comparison discussion and figures. Instead, we have added a short subsection with discussion about the orig-OH simulation using a 1-year simulation to more clearly make the point outlined above. By focusing on the coupled and uncoupled simulation results, we now clearly show the value of our improved simulation relative to the out-of-the-box simulations.**

2. **To simplify the analysis, we now focus the results on CO and CO$_2$ (and their chemical production terms) and remove parts of the CH$_4$ analysis.**

Here are some of my major concerns:

Table 1: For eaxmple, "Coupled only" : I do not understand - are all CO are produced from CH4 oxidation ? If so you are going to underestimate CO. If not, is CO in L(CO) and P(CO)ch4 are different entities, then there is a good chance of double counting

**Table 1 contains 3 sections:**

- **"Fields used by both uncoupled and coupled simulations"**
- **"Uncoupled only" – which we will change to "Fields used by uncoupled simulation only"**
- **"Coupled only" – which we will change to "Fields used by coupled simulation only"**

**The first section includes P(CO)$_{NMVOC}$, which is the other source of CO chemical production (CO produced from non-methane volatile organic compounds). We expect that by changing the table headings as described in the bullet points above it will become clearer that CH$_4$ oxidation is not the only source of CO chemical production.**

**Further, as stated in the caption and in line 87, Table 1 only provides the terms that impact the chemical production and loss fields (and vary between simulations). As stated in lines 87-88, the full list of all the flux terms (i.e., other emission or uptake fields) used by each simulation is presented in Table S1 in the Supplement (as these do not vary between the different simulations). In summary, all simulations include CO chemical production from CH$_4$, CO chemical production from NMVOCs, and direct CO emissions; hence there is no CO underestimation (excluding potential biases in the external emission fields). This is also shown in Equation (4), which shows that the simulations include CO emissions and total P(CO) chemical production, composed of the CH$_4$ and NMVOC terms as shown in Equation (5).**

**L(CO) and P(CO)$_{CH4}$ are entirely different terms. P(CO)$_{CH4}$ is defined in equation 5 (line 123) and represents the amount of CO produced when CH$_4$ reacts with OH (assuming a 100% CO yield from this reaction as explained on line 122). L(CO) represents the amount of CO lost via reaction with OH in the troposphere and from archived CO loss in the stratosphere. The L(CO) term is calculated at every model time step after all the emission and production fields (including P(CO)$_{CH4}$) are added to CO, hence there is no double counting. The calculated L(CO) (both troposphere and stratosphere) is then used to calculate the chemical production of CO$_2$. This simulation description is provided in detail in lines 110-141.**

Formulation of Eq. 1 & 2 (also for CO): Not correct !!, I think Eq. 1 and Eq. 2 are not separable in a chemistry-transport model, except for "tagging". Please clarify or rectify errors

**These equations are correct as written but were not adequately described in the text in the original manuscript. We now explicitly state in the revised manuscript that these equations only describe the changes in emission, deposition, production and loss terms that occur within each grid box and that advective transport fluxes between grid boxes (including between the troposphere and the stratosphere) are in addition to the terms described in each equation.**

**Note that the GEOS-Chem model dynamically calculates the tropopause height at every timestep, and uses this information (in all GEOS-Chem simulations) to assign each grid box to either the troposphere or the stratosphere. This happens before applying the appropriate source and sink terms as described in Equations (1) and (2) (and also for CO). We now clarify this in the main text.**

Figure 2: Quite large differences in OH. Acceptable? May be you should run CH3CCl3 tracer of checking your OH.

**We agree with the reviewer that there are large differences in OH values – which is one of the inconsistencies in the uncoupled simulations (currently used by the GEOS-Chem community) that our new coupled simulation is designed to eliminate.**

**To clarify, these different versions of the OH fields come from different historical full-chemistry simulations, which advanced from v5 (circa 2004) to v9 (circa 2013) due to developments and improvements to the full chemistry model. We try to highlight these differences and improve the simulation of all carbon gases by introducing the coupled simulation. Currently the GEOS-Chem community is using inconsistent OH fields across the uncoupled $CH_4$, CO and $CO_2$ simulations that will introduce biases in the modelled values. Introducing the new coupled simulation with consistent and more recently updated OH fields (as well the possibility to easily update the OH fields using future full chemistry model versions) eliminates these large differences. In our revised manuscript, we have removed the orig-OH simulation from the main discussion and reduced the discussion of the different OH fields. Where we do discuss them, we more clearly state that the v9 fields are the most up-to-date of the 3 fields used by the existing, out-of-the-box carbon gas simulations.**

**Although understanding and exploring the biases in the OH fields is an important task in atmospheric chemistry it is not the focus of our paper. The OH analysis in the paper is presented only for a better understanding of the changes in the chemical fields, and as such comparison with $CH_3CCl_3$ is beyond the scope of this work.**

Figure 5 (left column): the P(CO)ch4 and P(CO2) are apparently not consistent with the OH fields in Figure 2.

**If the reviewer is suggesting that changes between OH fields (Figure 2, strongest OH in v8 and weakest in v5) should be mirrored in the $P(CO)_{CH4}$ and $P(CO_2)$ chemical production fields, that is not correct.**

**While the differences in OH *contribute* to the differences between the simulations, they are not the only factor driving the variability in chemical production. For $P(CO)_{CH4}$, this term represents production of CO from $CH_4$ reaction with OH (equation 5) and therefore depends not only on [OH] but also on [$CH_4$], which differs between the uncoupled and coupled simulations. Lower OH values will not necessarily lead to a weaker chemical production: if the $CH_4$ amounts in the full chemistry simulation (used by the uncoupled simulation) are higher than the $CH_4$ initial fields in the coupled simulation, this can lead to overall stronger $P(CO)_{CH4}$ even if the OH levels are lower. We discuss these additional terms in detail in Section S1 in the Supplement. We now explain this relationship in the text when we discuss Figure 5. In addition, we have added text to Section 2 (model description) to describe the different methods used to calculate [$CH_4$] between the full chemistry (used for uncoupled) and coupled simulations.**

**For $P(CO_2)$, the situation is similar (the comparison between the uncoupled and coupled simulations will depend on both [OH] and [CO]). Here, however, we note that we only expect a very small change in $P(CO_2)$ between the coupled and coupled-origOH simulations. This is because, as shown in Figure 1, the ONLY difference between these simulations is the OH field that is used to calculate $P(CO)_{CH4}$. The latter represents only a fraction of the total CO source (which includes $P(CO)_{NMVOC}$ and direct CO emissions), and so this will translate to a smaller impact on $L(CO)=P(CO_2)$, consistent with the results shown in Figure 5. To state this another way, the v5 OH (turquoise) shown in Figure 2 is NOT used to calculate $L(CO)=P(CO_2)$ in the coupled-origOH simulation.**

**We recognise that this latter point is confusing and distracts from the main messages of our work. As stated previously, we have removed the coupled-origOH from the main body of the text, which we expect to make the comparison between the coupled and uncoupled simulations easier to understand. In addition, in the new subsection where we describe the coupled-origOH simulation, we more clearly explain that the v5 OH is ONLY used to calculate $L(CH_4) = P(CO)_{CH4}$ and not $L(CO) = P(CO_2)$.**

**Finally, we note that Figure 5 shows the total production summed over the full troposphere, while Figure 2 shows [OH] for specific model levels. We now state this explicitly in the caption to Figure 5.**

**In summary, we see no inconsistencies between Figures 2 and 5, and we expect that the changes outlined above will make this much clearer for readers. However, if we misunderstood the reviewer's comment we would appreciate further clarification as to what specifically is 'not consistent'.**

This where I had to stop going forward or read the text carefully. I am extremely sorry, but this has to be solved first before interpreting the results.
**As outlined above, there is no inconsistency to resolve. The revised manuscript has been reduced in complexity to make our aims, methods and results clearer.**

You have about 20% higher OH for the red and purple lines, compared to blue, both at the surface and at 500 mb when averaged over a year (Fig. 2).
But here in Fig. 5, we find the blue line is close to purple than the red line for P(CO2), and also I cannot explain the relative values of P(CO)ch4 as expected from the OH fields.
**Please refer to the detailed comments above. In particular, for $P(CO_2)$ the blue (coupled-origOH) and purple (coupled) lines SHOULD be closer to one another than to the red (uncoupled) line because they use the same OH for calculating $L(CO) = P(CO_2)$, and the only difference will come from the fraction of [CO] that originally derives from $P(CO)_{CH4}$.**

**As described above, we have made substantial modifications to the revised manuscript to make it easier to understand these subtleties, including removing the coupled-origOH simulation from the main text, better explaining these relationships where appropriate, and clarifying the differences in spatial scale between Figures 2 and 5.**

I understand that the OH level is affecting the concentrations of CO and CH4 and then you get very mixed pictures for P(CO) or P(CO2). But these are not realistic, because we only have one state of CO and CH4 concentration distributions (strictly).
**We agree with the reviewer that in the real atmosphere there is only one state of CO, $CH_4$, $CO_2$ and OH. Indeed, this was the main rationale for creating the coupled simulation, as in the uncoupled version there was a disconnect between these species that is not realistic. Our coupled simulation removes the disconnect, using only one consistent version of the OH and chemically linking all three carbon species distributions to one another.**

If you are checking the effect of OH then design experiments accordingly, and so on. Please consider.
**Please refer to our comments above. The focus of the paper is not to understand the effect of OH. We only presented the OH analysis to better explain the changes in the chemical production fields. As described in detail above, the modifications in the revised manuscript better streamline the aim and results of the paper.**

---

## Author Comment (AC3)

**In the text below we have included all the referee comments in black, followed by our response in red.**

**Anonymous Referee #3**

General Comments

The authors present coupled greenhouse gas simulations with GEOS-Chem focusing on $CO_2$/$CH_4$/CO and its global chemical interactions. Such simulations will be definitively needed towards a consistent description of long-term atmospheric chemistry and for realistic assessment of climatic change by earth system models. I fully acknowledge the work done here and I am convinced that the new developments implemented in GEOS-Chem are a big step towards these goals. When reading the abstract, I got the impression that the paper follows a clear outline by first comparing coupled and uncoupled simulations and then evaluate the new model version with observational data. The authors present a sound and informative introduction to the scientific and computational problem of consistent chemistry simulations, which comprise processes representing a broad range of timescales as well as trends and interannual variability. I also liked the detailed budget term quantification as presented in Table 2. Unfortunately, I got lost after different versions of the OH input fields were introduced. For me it remained unclear why the authors are not able to stick to a single OH field which then can be used for both, the uncoupled and the coupled simulations. In the discussion part of the manuscript, the introduction of a third simulation (coupled-origOH) lead to unnecessary confusion and an overload with information details which made it hard for me to extract the major conclusions. Overall, I would not recommend to publish the paper in its current form but I encourage the authors to submit a revised manuscript based on more consistent coupled and uncoupled simulations.

**We thank the reviewer for the comments and discussion. This comment (and those from the other reviewers) has highlighted to us that perhaps the manuscript did not clearly describe the aim of our work, and this has led to some confusion about the intent of our work and therefore the experimental design. We have substantially revised the manuscript to better clarify both of these points. Here, we summarise our rationale along with changes we have made in the manuscript.**

**There was a strong reason why the OH fields were different between simulations, as well as for introducing the coupled-origOH simulation. The aim of the work was to compare the *existing* GEOS-Chem $CH_4$, CO, and $CO_2$ simulations that are currently being used by the community to the improved, coupled simulation we have developed. What we wanted to demonstrate in the paper is that if one uses the uncoupled GEOS-Chem $CH_4$, CO, and $CO_2$ simulations out-of-the-box (which is how the GEOS-Chem community is using them), the chemical production fields (calculated using GEOS-Chem full chemistry simulations) are not consistent as they were created at different times using different versions of the full chemistry (and therefore different OH). Because the archived chemical fields used by the out-of-the-box uncoupled simulations were calculated with different OH fields, the resulting $CH_4$, CO, and $CO_2$ distributions were inconsistent with one another. The coupled simulation resolves these inconsistencies, which are partly driven by the lack of coupling and partly by the different existing chemical production fields. If we were to change to consistent OH fields between the uncoupled and coupled simulations, this would disguise the key limitations and biases in the chemical production/loss in the uncoupled simulations. It would also be of limited value, as we would effectively be creating a false narrative: comparing our new coupled simulation to individual $CH_4$, CO, and $CO_2$ simulations that do not exist and are not being used by the community.**

**One thing that the comparison between the uncoupled (out-of-the-box) and coupled simulations does not highlight is inconsistencies between the out-of-the-box $CH_4$ and CO simulations. The out-of-the-box uncoupled $CH_4$ simulation uses v5 OH, while the out-of-the-box uncoupled CO simulations uses $P(CO)_{CH4}$ based on a full chemistry simulation with v9 OH. This means that in the uncoupled simulations, $L(CH_4)$ and $P(CO)_{CH4}$ are *not* equivalent (although, in reality, they should be). The reason for introducing the coupled-origOH simulation was to demonstrate the impact of this discrepancy, by determining what the coupled simulation would look like if we calculated $P(CO)_{CH4}$ (and follow-on parameters) from the same $L(CH_4)$ as in the out-of-the-box $CH_4$ simulation. We understand that this is a fairly subtle point, and that its inclusion alongside the coupled and uncoupled simulations has hampered the readability of the manuscript.**

**To improve the paper, we have made a number of changes in the revised manuscript. In addition to clarifying the above points throughout, we have also made the following substantial changes:**

1. **We focus the analysis on comparison of the uncoupled and coupled simulations only, removing the orig-OH simulation from the main comparison discussion and figures. Instead, we have added a short subsection with discussion about the orig-OH simulation using a 1-year simulation to more clearly make the point outlined above. By focusing on the coupled and uncoupled simulation results, we now clearly show the value of our improved simulation relative to the out-of-the-box simulations.**
2. **To simplify the analysis, we now focus the results on CO and $CO_2$ (and their chemical production terms) and remove parts of the $CH_4$ analysis.**

**We would kindly ask Reviewer #3 to also visit our detailed reply to Reviewer #1's general comment about the experimental setup for further explanation of these points. We anticipate that the improvements we introduce in the revised manuscript will much better highlight and clarify the aim of our work and the reasoning behind our experimental design.**

Specific Comments

Line 49: What do you mean by "outside source regions"?
**"outside source regions" was referring to areas/regions that are not dominated by strong anthropogenic point emissions. We clarify this in the revised manuscript.**

Lines 49-56: For CO budget terms you can also refer to Stein et al. (2014).
**We thank the reviewer for suggesting the additional reference, which we have added to the revised manuscript (both the text and Table 2).**

Line 51: Publication year is missing.
**Fixed.**

Lines 52-54: It would be interesting to see also the numbers for the chemical production by NMVOCs (used as input for your simulations).
**We now state the P(NMVOC) numbers in the revised manuscript. On lines 52-54 we list known literature values for this chemical production with references (320–820 Tg CO yr$^{-1}$, Holloway et al., 2000; Bergamaschi et al., 2000; Arellano Jr. and Hess, 2006; Duncan et al., 2007; Zeng et al., 2015; Fisher et al., 2017) while in Section 2.2 (coupled simulation description) we present the numbers used as input in our simulations (480 Tg CO yr$^{-1}$).**

Lines 60-63: Can you give a reference here?
**We have added references in the revised manuscript that highlight differences in the chemical**

**fields: Nassar et al., (2010), Fisher et al., (2017),  Duncan et al., (2007), Suntharalingam et al., (2005), Zeng et al., (2015).**

Lines 82-84: Is the spin-up time sufficient? You doubt this later on (Lines 506-508).
**Our spin up period totals to 11 years, and we used this time period based on the recommendation from the GEOS-Chem team. The recommended spin up period for $CO_2$ and $CH_4$ is 10 years ([http://wiki.seas.harvard.edu/geos-chem/index.php/GEOS-Chem_restart_files](http://wiki.seas.harvard.edu/geos-chem/index.php/GEOS-Chem_restart_files)). We clarify this in the revised manuscript, and have removed the comment on lines 507-508 referring to insufficient spin-up time.**

Lines 96-97: I would expect that a single reference full chemistry simulation is used for all simulations presented here.
**Please refer to our response to the general comment above. In addition, we now clarify in Section 2.1 that the uncoupled simulations run here are the out-of-the-box versions available in GEOS-Chem v12.1.1, which were developed independently at different points in time and therefore use chemical production fields that come from different full chemistry simulations. We better highlight in the revised text that these discrepancies between full-chemistry versions are a major source of bias (and limitation) in the uncoupled $CH_4$, CO and $CO_2$ simulations; and that this significant limitation is now removed by introducing the coupled simulation.**

Line 134: How does GEOS-Chem handle Biomass burning emissions? It is known that such emissions need to be emitted throughout the troposphere following a vertical profile.
**As our intention is to understand the impact of the coupling (including consistent production and loss fields) relative to the out-of-the-box uncoupled simulation, we treat emissions from biomass burning identically between the coupled and uncoupled simulation, using the treatment found in the out-of-the-box uncoupled simulations. However, for $CO_2$ we made a small modification by updating the simulation to use QFEDv2 emissions that were not available in the out-of-the-box uncoupled $CO_2$ simulation. We implemented this following the method used in the $CH_4$ simulation and we did this to have consistent biomass burning emission types across all three species. We now describe the biomass burning emissions treatment in section 2.2. Briefly: for all three gases we use daily QFEDv2 emissions (Table S1, Supplement) with additional diurnal scale factors. For CO we use vertical partitioning of the emissions where 35% of the biomass burning emissions are emitted above the Planetary Boundary Layer (again, following the default setup in the uncoupled CO simulation). For $CO_2$ and $CH_4$, the emissions are emitted at the surface only and transported to higher altitudes via mixing. We acknowledge that the $CH_4$ and $CO_2$ emissions treatment should be improved to match that for CO; however, for this work we aimed to keep all aspects of the simulations not associated with the chemical coupling as consistent as possible with the default version of the uncoupled simulations.**

Line 295: Exchange "tropospheric column" by "mid troposphere".
**Fixed; however, this part of the discussion will be moved into the Supplement based on the comments from Reviewer #1.**

Lines 348-349: I would expect to have the same P(NMVOC) for all model runs. Give numbers!
**P(NMVOC) is consistent in the simulations where it is used as an input field: uncoupled CO and coupled simulation. In these simulations the P(NMVOC) is based on v9 full chemistry simulations and additional offline processing as described in Fisher et al., (2017). However, in the uncoupled $CO_2$ simulation, the P($CO_2$) chemical fields were archived by Nassar et al., (2010) using the v8 full chemistry simulation (with differences in chemical scheme that would implicitly include differences in P(NMVOC)). Intermediate terms, including P(NMVOC), were not archived, limiting us from specifying the numbers in the revised manuscript.**

Table 1: Publication year is missing.
**Fixed.**

Figure 1: I like this figure. It could even improve if you orient your coupled and uncoupled flows from left to right.
**Figure modified as suggested.**

Figure 4: From your description I would expect that P(CO)$_{CH4}$ (for all years) and P(CO$_2$) (for 2010-2017) remains exactly constant for the uncoupled runs (except for leap years).
**That is correct. The chemical production input fields have no inter-annual variability in the case of CO and after year 2010 for CO$_2$; however, Figure 4 does show a change. The reason for this is: 1) as the reviewer noted leap years during 2008, 2012 and 2016 that will lead to higher production (since the figure shows total production summed over the year) and 2) interannual variability in the meteorological fields (e.g., pressure levels, tropopause height) affecting the calculation of the total tropospheric budget. We clarify this in the revised manuscript.**

Figure 7(b): This reads like "Surface Loss" as parameter.
**Fixed, however, this Figure will be moved into the Supplement based on comments from Reviewer #1.**

**References:**

Stein, O., Schultz, M. G., Bouarar, I., Clark, H., Huijnen, V., Gaudel, A., George, M., and Clerbaux, C.: On the wintertime low bias of Northern Hemisphere carbon monoxide found in global model simulations, Atmos. Chem. Phys., 14, 9295-9316, doi:10.5194/acp-14-9295-2014, 2014.

**Holloway, T., Levy II, H., and Kasibhatla, P.: Global distribution of carbon monoxide, Journal of Geophysical Research: Atmospheres, 105, 12 123–12 147, https://doi.org/10.1029/1999JD901173, 2000.**

**Bergamaschi, P., Hein, R., Brenninkmeijer, C. A. M., and Crutzen, P. J.: Inverse modeling of the global CO cycle: 2. Inversion of 13C/12C and 18O/16O isotope ratios, Journal of Geophysical Research: Atmospheres, 105, 1929–1945, https://doi.org/10.1029/1999JD900819, 2000.**

**Arellano Jr., A. F. and Hess, P. G.: Sensitivity of top-down estimates of CO sources to GCTM transport, Geophysical Research Letters, 33, https://doi.org/10.1029/2006GL027371, 2006.**

**Duncan, B. N., Logan, J. A., Bey, I., Megretskaia, I. A., Yantosca, R. M., Novelli, P. C., Jones, N. B., and Rinsland, C. P.: Global budget of CO, 1988–1997: Source estimates and validation with a global model, Journal of Geophysical Research: Atmospheres, 112, https://doi.org/10.1029/2007JD008459, 2007**

**Zeng, G., Williams, J. E., Fisher, J. A., Emmons, L. K., Jones, N. B., Morgenstern, O., Robinson, J., Smale, D., Paton-Walsh, C., and Griffith, D. W. T.: Multi-model simulation of CO and HCHO in the Southern Hemisphere: comparison with observations and impact of biogenic emissions, Atmospheric Chemistry and Physics, 15, 7217–7245, https://doi.org/10.5194/acp-15-7217-2015, 2015.**

**Fisher, J. A., Murray, L. T., Jones, D. B. A., and Deutscher, N. M.: Improved method for linear carbon monoxide simulation and source attribution in atmospheric chemistry models illustrated**

using GEOS-Chem v9, Geoscientific Model Development Discussions, 2017, 1–24, https://doi.org/10.5194/gmd-2017-94, 2017.

Nassar, R., Jones, D. B. A., Suntharalingam, P., Chen, J. M., Andres, R. J., Wecht, K. J., Yantosca, R. M., Kulawik, S. S., Bowman, K. W., 765 Worden, J. R., Machida, T., and Matsueda, H.: Modeling global atmospheric $CO_2$ with improved emission inventories and $CO_2$ production from the oxidation of other carbon species, Geoscientific Model Development, 3, 689–716, https://doi.org/10.5194/gmd-3-689-2010, 2010.

Suntharalingam, P., Randerson, J. T., Krakauer, N., Logan, J. A., and Jacob, D. J.: Influence of reduced carbon emissions and oxidation on the distribution of atmospheric $CO_2$: Implications for inversion analyses, Global Biogeochemical Cycles, 19, https://doi.org/10.1029/2005GB002466, 2005.